# Are We Merely Justifying Results ex Post Facto? Quantifying Explanatory Inversion in Post-Hoc Model Explanations

## Abstract

Post-hoc explanation methods provide interpretation by attributing predictions to input features. Natural explanations are expected to interpret how the inputs lead to the predictions. Thus, a fundamental question arises: *Do these explanations unintentionally reverse the natural relationship between inputs and outputs?* Specifically, are the explanations rationalizing predictions from the output rather than reflecting the true decision process? To investigate such *explanatory inversion*, we propose *Inversion Quantification* (*IQ*), a framework that quantifies the degree to which explanations rely on outputs and deviate from faithful input-output relationships. Using the framework, we demonstrate on synthetic datasets that widely used methods such as LIME and SHAP are prone to such inversion, particularly in the presence of spurious correlations, across **tabular**, **image**, and **text** domains. Finally, we propose *Reproduce-by-Poking* (`RBP`), a simple and model-agnostic enhancement to post-hoc explanation methods that integrates forward perturbation checks. We further show that under the IQ framework, `RBP` theoretically guarantees the mitigation of explanatory inversion. Empirically, for example, on the synthesized data, `RBP` consistently reduces the inversion across iconic post-hoc explanation approaches and domains.

## 1 Introduction

Post-hoc explanation methods have become essential tools for interpreting the predictions of complex machine learning (ML) models, particularly in high-stakes applications such as healthcare (Turbé et al., 2023), finance (de-la Rica-Escudero et al., 2025), and policy-making (Heesen et al., 2024). By providing insights into which input features are most influential for a given output, methods such as SHAP (Lundberg, 2017), LIME (Ribeiro et al., 2016), and Integrated Gradients (Sundararajan et al., 2017) aim to increase the transparency of predictive models. Post-hoc explanation methods provide interpretation by attributing predictions to input features, while natural explanations are expected to interpret how the inputs result in the predictions. Therefore, a critical question arises:

> **RQ**: *Do post-hoc explanations faithfully represent the model's decision-making process, without inadvertently rationalizing predictions from the output?*

This potential reversal of reasoning, which we term *explanatory inversion*, undermines the reliability of explanations and poses challenges to the broader adoption of AI systems in sensitive domains.

As illustrated in Figure 1, explanatory inversion arises when a post-hoc explanation method over-relies on the model's output in generating attributions, rather than accurately reflecting the relationship between inputs and predictions. For instance, a method may highlight features that appear important solely because they correlate with the model's output, rather than because they genuinely influence the prediction. We further theoretically show that the alignment of explanations with ground truth is inversely bounded by the degree of explanatory inversion. Thus, quantifying such inversion is essential for identifying the limitations of existing explanation techniques and for developing more robust alternatives. To investigate and mitigate explanatory inversion, we make these contributions:

**(1) Formal Definition and Framework:** We formally define explanatory inversion and introduce the *Inversion Quatification (IQ)*, a novel framework that quantifies the degree to which explanations rely on outputs and deviate from faithful input-output relationships. IQ evaluates explanations along two key dimensions: *reliance on outputs*, measuring the correlation between attributions and model predictions, and *faithfulness*, assessing alignment with perturbations of input features. IQ is applicable to real-world data without ground-truth input importance. **(2) Empirical Validation:** Using synthetic datasets across tabular, image, and text domains, we systematically verify the **presence** of explanatory inversion in widely used methods such as LIME and SHAP. Our experiments reveal that these methods are particularly vulnerable to spurious correlations, leading to significant explanatory inversion. **(3) Mitigation via Reproduce-by-Poking:** We propose *Reproduce-by-Poking* (RBP), a simple and model-agnostic enhancement to post-hoc expla-

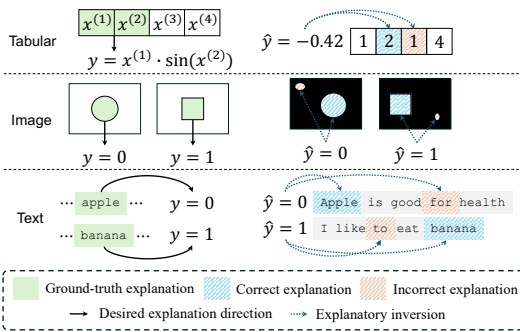

Figure 1: Illustration of post-hoc explanation methods and the potential of explanatory inversion. For tabular data (first row), ground-truth explanations attribute the output $y = x^{(1)} \cdot sin(x^{(2)})$ to the contributions of features $x^{(1)}$ and $x^{(2)}$, but explanatory inversion misattributes $x^{(2)}$ and $x^{(3)}$. For image data (second row), explanations should focus on the correct object, but explanatory inversion leads to incorrect focus regions. For text data (third row), ground-truth explanations link keywords to labels, but explanatory inversion results in misaligned or irrelevant attributions.

nation methods that incorporates forward perturbation checks. Under the IQ framework, we demonstrate that RBP provably mitigates explanatory inversion. For example, on synthesized data, RBP reduces inversion by **1.8%** on average across iconic post-hoc explanation approaches and domains.

## 2 RELATED WORK AND PRELIMINARIES

**Post-hoc Explanation Methods.** Post-hoc techniques have become indispensable for interpreting predictions of complex models. SHAP (Lundberg, 2017) leverages game-theoretic principles to ensure attributions satisfy desirable properties such as additivity and consistency. LIME (Ribeiro et al., 2016) approximates local model behavior using interpretable surrogate models, while Integrated Gradients (Sundararajan et al., 2017) integrates gradients along a path from a baseline to the input, ensuring completeness and sensitivity. Other methods include Occlusion (Zeiler and Fergus, 2014), which computes feature importance by masking input regions, and SmoothGrad (Smilkov et al., 2017), which reduces noise in saliency maps via input perturbations. Post-hoc explanation techniques are widely adopted but are shown to rely heavily on model outputs, based on our results, raising concerns about their susceptibility to explanatory inversion (Rodis et al., 2024). Recent advancements in post-hoc explanation methods have led to the development of more techniques to handle specific tasks (Turbé et al., 2023; Leemann et al., 2023). See a more detailed description in Appendix B.

**Evaluating Explanation Reliability.**

Ensuring the reliability of explanations is a critical area of research. Adebayo et al. (2018) demonstrate that many explanation methods fail basic sanity checks. Faithfulness is often evaluated using perturbation-based approaches; for instance, Hooker et al. (2019) introduce removal-based benchmarks measuring performance degradation when important features are removed. Wang et al. (2020a) examine how explanations can be manipulated to justify specific outcomes. Dimanov et al. (2020) analyze fairness in post-hoc explanations, showing that feature importance methods can perpetuate biases present in the data. Metrics such as infidelity assess how well explanations capture the model's local behavior under input perturbations, while sensitivity measures how much explanations change with small input variations (Yeh et al., 2019). The robustness or stability of explanations against input changes is another focus (Alvarez-Melis and Jaakkola, 2018), with recent work exploring potential trade-offs between robustness and faithfulness, particularly in contexts like adversarial training (Tan and Tian, 2023). Mohseni et al. (2021) reviewed challenges and benchmarks in this area. Differing from them, this paper identifies an orthogonal problem of post-hoc explanations that they can excessively rely on model outputs instead of the inherent decision-making process, especially when spurious correlation exists (Alvarez-Melis and Jaakkola, 2018; Slack et al., 2020). We give a more detailed discussion on the difference with existing metrics in Appendix C.

**Preliminaries: Post-hoc Explanation.** Post-hoc explanation methods provide interpretations of model predictions by assigning importance scores to input features. Our analysis in this paper centers on four widely adopted and domain-general techniques: SHAP (SHapley Additive exPlanations) (Lundberg, 2017), LIME (Local Interpretable Model-Agnostic Explanations) (Ribeiro et al., 2016), Integrated Gradients (IG) (Sundararajan et al., 2017), and Occlusion (Zeiler and Fergus, 2014). These methods were selected due to their prevalence and broad applicability across various models and data types, allowing for consistent evaluation. While numerous other explanation techniques, including domain-specific ones, exist (e.g., GradCAM (Selvaraju et al., 2017)), our focus remains on these general approaches. Their detailed formal definitions are deferred to Appendix A.

## 3 EXPLANATORY INVERSION QUANTIFICATION (IQ)

In this section, we present the theoretical foundations of the proposed *Inversion Quantification* framework for evaluating post-hoc explanations. We first formalize explanatory inversion and its implications for model interpretability (§3.1). Then, we derive the components of IQ (§3.2) and establish its applicability in both synthetic and real-world contexts.

### 3.1 EXPLANATORY INVERSION

Let $\mathbf{x} \in \mathbb{R}^d$ denote the input vector, $M : \mathbb{R}^d \to \mathbb{R}^k$ represent a machine learning model mapping inputs to outputs, and $\mathcal{E} : \mathbb{R}^d \to \mathbb{R}^d$ denote a post-hoc explanation method assigning attributions $\mathbf{a} = \mathcal{E}(\mathbf{x})$. A reliable explanation should reflect the forward relationship between $\mathbf{x}$ and $M(\mathbf{x})$. However, explanatory inversion occurs when the attributions $\mathbf{a}$ primarily rely on the output $M(\mathbf{x})$, rather than capturing the input-output relationship.

**Definition 3.1. [Explanatory Inversion]** We define *explanatory inversion* as the degree to which an explanation method $\mathcal{E}$ depends on the model's output $M(\mathbf{x})$ to generate attributions. The attributions $\mathbf{a}$ are expressed as:

$$\mathbf{a} = f_{\mathcal{E}}(M(\mathbf{x}), \mathbf{x}), \tag{1}$$

where $f_{\mathcal{E}}$ is the internal mechanism of the explanation method. Explanatory inversion is maximal when $\mathbf{a}$ is independent of $\mathbf{x}$, i.e., $\|\partial \mathbf{a}/\partial \mathbf{x}\| \approx 0$, and minimal when $\|\partial \mathbf{a}/\partial M(\mathbf{x})\| \approx 0$.

### 3.2 INVERSION QUANTIFICATION

Inversion Quantification (IQ) evaluates explanations along two key dimensions: reliance on outputs and explanation faithfulness. Intuitively, if the explanation does not rely on the outputs and aligns well with the model's decision-making process, this explanation is considered less likely to involve explanatory inversion. In the following, we first quantify the reliance on outputs ($R$) and the explanation faithfulness $F$, which are closely related to explanatory inversion. After that, we define the Inversion Score (IS) as a metric for IQ, combining $R$ and $F$.

**Definition 3.2. [Reliance on Outputs, $R$]** The reliance score quantifies the degree to which the attributions $\mathbf{a}$ are influenced by the model's output $M(\mathbf{x})$, rather than the input-output relationship. For a dataset $\mathcal{D} = \{(\mathbf{x}_i, M(\mathbf{x}_i))\}_{i=1}^{N}$, $R$ is defined as:

$$R = \frac{1}{N} \sum_{i=1}^{N} \frac{1}{d} \sum_{j=1}^{d} \rho\big(\Delta a_i^{(j)}, \Delta M(\mathbf{x}_i; j)\big), \tag{2}$$

where $d$ is the number of features, $\rho$ is the correlation coefficient, $\Delta a_i^{(j)} = a_i^{(j)} - a_{\text{base},i}^{(j)}$ represents the change in the attribution for feature $j$ after perturbation, where $a_{\text{base},i}^{(j)}$ is the baseline attribution for feature $j$, and $\Delta M(\mathbf{x}_i; j)$ is the observed change in the model's output due to perturbing feature $j$ in input $\mathbf{x}_i$. A higher $R$ means stronger reliance on $M(\mathbf{x})$ and less explanatory inversion, vice versa.

**Definition 3.3. [Explanation Faithfulness, $F$]** Faithfulness evaluates how well the attributions $\mathbf{a}$ align with the actual effect of features on the model's output. For a dataset $\mathcal{D} = \{(\mathbf{x}_i, M(\mathbf{x}_i))\}_{i=1}^{N}$, $F$ is defined as:

$$F = \frac{1}{N} \sum_{i=1}^{N} \frac{\sum_{j=1}^{d} a_i^{(j)} \big| M(\mathbf{x}_i) - M(\mathbf{x}_i^{(-j)}) \big|}{\sum_{j=1}^{d} |a_i^{(j)}|}, \tag{3}$$

where $a_i^{(j)}$ is the attribution assigned to feature $j$ for sample $\mathbf{x}_i$, and $M(\mathbf{x}_i^{(-j)})$ represents the model's output with feature $j$ perturbed or masked.

According to the definition, a higher $F$ indicates that the attributions more faithfully reflect the effect of features on the model's output, thus relatively suffering less from explanatory inversion. Based on the definitions of $R$ and $F$, we define the inversion score (IS) as follows:

**Definition 3.4. [Inversion Score, $IS$]** The Inversion Score (IS) quantifies the extent of explanatory inversion by combining Reliance on Outputs ($R$) and Explanation Faithfulness ($F$) into a single metric. IS is defined as:

$$\text{IS}(R, F) = \left( \frac{R^p + (1 - F)^p}{2} \right)^{\frac{1}{p}}, \tag{4}$$

where $p$ is a hyperparameter controlling the sensitivity to deviations in $R$ and $F$. By default, $p = 2$ yields a quadratic mean IS $\in [0, 1]$, which balances the contributions of $R$ and $F$. An ideal explanation with $R = 0, F = 1$ will lead to IS $= 0$. An explanation purely based on inversion will have $R = 1, F = 0$ will lead to IS $= 1$.

**Justification.** The power mean with $p > 1$ emphasizes larger deviations in either $R$ or $(1 - F)$, ensuring that significant shortcomings in one dimension dominate more for the score. This design reflects the intuition that both low faithfulness and high reliance on outputs severely undermine the quality of explanations, and their combined effect should be penalized. The choice of $p$ provides flexibility to adjust this sensitivity to the application.

To justify the effectiveness of our proposed inversion scores measuring explanatory inversion, we prove the theorem:

**Theorem 3.5.** *The proposed Inversion Score (IS) effectively quantifies explanatory inversion as defined in Definition 3.1. The proof is presented in Appendix E.1. Specifically:*
1. *Higher IS indicates stronger reliance on the model's output $M(\mathbf{x})$ (captured by $R$) and weaker alignment of attributions $\mathbf{a}$ with the causal effects of features on $M(\mathbf{x})$ (captured by $F$).*
2. *Lower IS reflects explanations that minimize reliance on $M(\mathbf{x})$ and maximize faithfulness to the forward relationship between $\mathbf{x}$ and $M(\mathbf{x})$.*

According to Theorem 3.5, RBP can effectively measure the degree of explanatory inversion presented in explanations. In addition, in scenarios where ground-truth feature importance $\mathbf{g}$ is available (e.g., synthetic datasets), we define the alignment score $A$ as follows:

**Definition 3.6. [Explanation Alignment, $A$]** Given the attribution vector $\mathbf{a}$ and the ground-truth vector $\mathbf{g}$, the explanation alignment $A$ is calculated as their cosine similarity:

$$A = \frac{\langle \mathbf{a}, \mathbf{g} \rangle}{\|\mathbf{a}\| \|\mathbf{g}\|}. \tag{5}$$

This definition applies across tabular, image, and text data by adapting $\mathbf{g}$ appropriately (e.g., causal features, bounding boxes, or ground-truth tokens).

Based on the definition of explanation alignment, we further derive the following theorem that connects the explanation's similarity to ground truth with its inversion score.

**Theorem 3.7.** *For any explanation method $\mathcal{E}$ and a model $M$, if the ground-truth explanation $\mathbf{g}$ is available, then the explanation alignment score $A$ serves as an upper bound on explanation quality with respect to Inversion Score $IS$. The proof is given in Appendix E.2. Specifically,*

$$A \leq 1 - \gamma \cdot \text{IS}(R, F), \tag{6}$$

*for some constant $\gamma > 0$ that depends on the sensitivity of $A$ to attribution misalignment.*

**Interpretation.** This theorem indicates that higher explanatory inversion, as captured by the Inversion Score $IS$, implies a lower upper bound on alignment with ground truth. Intuitively, as attributions deviate from being faithful and increasingly rely on model outputs, their alignment with causally valid ground truth diminishes.

### 3.3 SPURIOUS FEATURE INJECTION

As shown in figure 3, to evaluate the robustness of explanation methods against explanatory inversion, we introduce a spurious feature $\widetilde{x}_{\text{spur}}$ that is correlated with the model's output. This feature is injected **only during inference time**, ensuring that the decision-making learned by the model during training remains unchanged. $\widetilde{x}_{\text{spur}}$ is defined as:

$$\widetilde{x}_{\text{spur}} = \psi M(\mathbf{x}) + \varepsilon, \quad \varepsilon \sim \mathcal{N}(0, \sigma^2), \tag{7}$$

Figure 2: Visualization of feature attributions for the shape classification task under both normal and spurious scenarios. Each row displays an input image and feature attributions generated by four post-hoc explanation methods: Integrated Gradients (IG), Occlusion, Shapley Value Sampling, and LIME. Columns show comparisons between normal (left) and spurious (right) conditions. The spurious scenario introduces a bright distractor pixel in the top-left corner of images labeled as 1, which leads to incorrect attributions in several methods. Desired focus regions (e.g., the object shape) are highlighted under normal conditions, while spurious conditions shift the attributions toward the irrelevant injected pixel. More case studies are included in Appendix K.

where $\psi$ controls the strength of the correlation between $\widetilde{x}_{\mathrm{spur}}$ and the model's output $M(\mathbf{x})$, and $\varepsilon$ is the random noise.

**Motivation.** By design, this injection ensures that the model's decision boundary and predictions remain unaffected, but it provides a new feature that is heavily correlated with the prediction, and explanation methods may erroneously attribute importance to this new feature. A good post-hoc explanation method should maintain the original explanation and not assign undue importance to the spurious feature. Any changes in the explanation reflect the degree of explanatory inversion. The impact of the spurious feature on explanation methods is assessed by measuring the difference in the Inversion Score (IS) before and after introducing $\widetilde{x}_{\mathrm{spur}}$, denoted as $\Delta\mathrm{IS} = \mathrm{IS}_{\mathrm{spur}} - \mathrm{IS}_{\mathrm{base}}$, where $\mathrm{IS}_{\mathrm{base}}$ is the IS computed on the original dataset. $\mathrm{IS}_{\mathrm{spur}}$ is the IS computed after injecting the spurious feature. A larger $\Delta\mathrm{IS}$ means the explanation method is more susceptible to spurious correlations, showing its vulnerability to explanatory inversion.

Figure 3: Illustration of IQ with spurious feature injection and its impact on explanations across modalities. For tabular data (first row), during training, feature $x^{(3)}$ follows a standard normal distribution, making it independent of the target variable. At test time, a spurious correlation is introduced where $x^{(3)}$ is linearly dependent on $y$ with noise $\varepsilon$, leading to incorrect reliance. For image data (second row), a distractor is injected into the test set, shifting explanations toward irrelevant regions. For text data (third row), an additional token (e.g., `peach`) appears in test samples, causing explanations to assign importance to non-informative words.

## 4 REPRODUCE-BY-POKING (RBP)

To address the issue of explanatory inversion, we propose *Reproduce-by-Poking (RBP)*, a novel enhancement to post-hoc explanation methods. RBP incorporates forward perturbation checks into the attribution process, encouraging that attributions reflect genuine input-output relationships rather than artifacts of model outputs. This section describes the design of RBP and provides an intuitive and mathematical justification for its effectiveness.

### 4.1 FRAMEWORK OF RBP

RBP refines standard post-hoc explanations through forward perturbation checks, introducing an additional validation step. First, we compute the baseline attributions, denoted as $\mathbf{a}$, using any established explanation method $\mathcal{E}$ (e.g., SHAP, LIME, IG, or Occlusion). These baseline attributions reflect the initial assessment of how each feature in the input $\mathbf{x}$ influences the model's output.

**Forward Perturbation Checks.** For each sample $\mathbf{x}$ and its feature $j$, RBP perturbs the feature multiple times to generate a series of modified inputs, $\mathbf{x}_{\mathrm{pert},j}$. The goal is to simulate slight variations in the feature values while maintaining an unaltered prediction, i.e., $M(\mathbf{x}_{\mathrm{pert},j}) = M(\mathbf{x})$. At each perturbation, new attributions, $\mathbf{a}_{\mathrm{pert},j}$, are computed. The deviation for feature $j$ is then quantified as:

$$\delta^{(j)} = \frac{1}{n_{\mathrm{pert}}} \sum_{k=1}^{n_{\mathrm{pert}}} |a_{\mathrm{pert}(k)}^{(j)} - a^{(j)}|, \tag{8}$$

where $a^{(j)}_{\text{pert}(k)}$ represents the attribution for feature $j$ after the $k$-th perturbation, and $a^{(j)}$ is the baseline attribution. This step captures the stability of the attributions under small, controlled changes to the input. Features with high deviations are indicative of attributions that are overly sensitive to minor perturbations, suggesting potential unreliability.

**Adjust Baseline Attributions.** In the final step, RBP refines the initial attributions by penalizing features with high deviations. The refined attributions, $\tilde{a}$, are computed as:

$$\tilde{a}^{(j)} = \frac{a^{(j)}}{1 + \delta^{(j)} \cdot \lambda}, \qquad (9)$$

where $\lambda > 0$ is a hyperparameter controlling the impact of the deviation on the adjustment. This adjustment reduces the influence of features whose attributions are unstable, which encourages the refined attributions to be more robust.

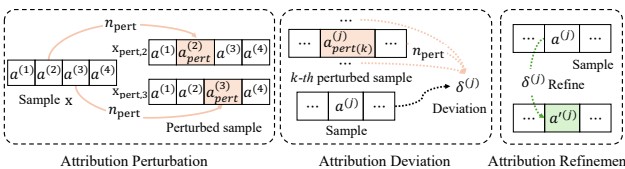

Figure 4: Overview of RBP, divided into three stages. In the *Attribution Perturbation* stage (left), multiple perturbed samples are generated from a given input sample $\mathbf{x}$, altering feature values. In the *Attribution Deviation* stage (middle), deviations $\delta^{(j)}$ are computed for each feature $a^{(j)}$ based on differences across perturbations. Finally, in the *Attribution Refinement* stage (right), attributions are refined by reducing the influence of features with high deviation, yielding adjusted attributions $a'^{(j)}$.

**Intuition** behind forward perturbation in RBP is that reliable attributions should remain stable under small, localized changes to input features. If the attribution for a feature fluctuates significantly despite the model's prediction staying constant, it indicates that the attribution is overly sensitive and may not reflect the true causal importance of the feature. Conversely, stable attributions under such perturbations suggest that the feature's importance is accurately captured, aligning with the model's actual behavior. By penalizing unstable attributions, RBP ensures explanations are more robust and faithful to the model's underlying mechanisms.

## 4.2 THEORETICAL ANALYSIS

We present key theoretical properties of RBP that demonstrate its ability to reduce explanatory inversion, and a theorem to indicate the robustness of RBP against spurious features. Proofs of the theorems are in Appendix E.

**Theorem 4.1** (Improved Reliance and Faithfulness with RBP). *The RBP method achieves both a reduction in output reliance ($R_{RBP}$) and an improvement in faithfulness ($F_{RBP}$) compared to the baseline scores ($R$ and $F$, respectively):*

$$R_{RBP} < R \qquad and \qquad F_{RBP} > F \qquad (10)$$

*The reduction in reliance indicates a decrease in explanatory inversion by penalizing attributions that fail to align with forward perturbations. The improvement in faithfulness indicates better alignment of attributions with true feature effects through perturbation-driven adjustments.*

Theorm 4.1 demonstrates that RBP is able to reduce explanatory inversion by reducing reliance on output and improving explanation faithfulness. To demonstrate the performance of RBP in the presence of spurious features, we further prove the following theorem:

**Theorem 4.2** (Resilience to Spurious Features). *Let $\widetilde{x}_{spur}$ be a spurious feature introduced during inference according to equation 7. If $\widetilde{x}_{spur}$ does not causally influence $M(\mathbf{x})$, RBP ensures that the adjusted attribution $\tilde{\mathbf{a}}_{spur}$ for the spurious feature converges to zero:*

$$\tilde{\mathbf{a}}_{spur} \to 0 \quad if\ \delta_{spur} \gg 0\ or\ \Delta M(\mathbf{x}; \widetilde{x}_{spur}) \approx 0. \qquad (11)$$

According to Theorem 4.2, RBP is able to produce adjusted attribution that converges to zero when the feature is spurious. In summary, RBP mitigates explanatory inversion by grounding attributions in forward perturbations, reducing reliance on the model's outputs and increasing the overall reliability of post-hoc explanations.

## 5 EXPERIMENTS

Our evaluation spans multiple **modalities**, *i.e.*, tabular, image, and text, under controlled settings where artificially introduced *spurious features* may influence post-hoc explanations. We consider both regression and classification **tasks**, and cover a range of **models**, including traditional machine learning methods (*e.g.*, SVM and random forests) and deep learning architectures (*e.g.*, multilayer perceptrons (MLPs), convolutional neural networks (CNNs), and transformers).

## 5.1 EXPERIMENTAL SETUP

Each dataset consists of a training set and two test sets: a **standard** test set and a **spurious** test set, where an additional irrelevant feature is introduced at inference time. The evaluation metrics include *reliance on outputs* ($R$), *faithfulness* ($F$), the overall *Inversion Score* ($IS$), the change in IS due to spurious features ($\Delta IS$), and *alignment* ($A$). To ensure fair comparisons, we confirm that all models converge during training and achieve high predictive performance on the original test set (see Appendix F, G for implementation details). This ensures that models correctly capture the intended data relationships, allowing us to isolate the effects of spurious feature injection on explanation methods. Experimental results on real-world datasets are in Appendix L.

▷ **Synthetic Tabular Data**: Multi-Feature Regression. We construct a synthetic regression dataset with six input features: $\{x^{(1)}, x^{(2)}\}$ are informative, while $\{x^{(3)}, x^{(4)}, x^{(5)},$ $x^{(6)}\}$ serve as distractors. The datasets are generated according to the following randomly chosen non-linear function. Similar patterns are observed with other functions.

$$y = x^{(1)} \cdot \sin(x^{(1)}) \cdot \log(1 + |x^{(2)}|) + \varepsilon. \tag{12}$$

**Spurious Feature Injection.** In the spurious test set, we transform one of the dummy features by $\tilde{x}^{(j)} = \psi y + \epsilon, (j \in \{3, 4, 5, 6\})$, which is correlated with the target variable at test time but is not used for learning during training. This simulates a setting where models may inadvertently rely on spurious correlations in explanations.

**Models.** We evaluate a range of regression models, including: Random Forest Regressor, Support Vector Regression (with RBF kernel), and MLP.

▷ **Synthetic Image Data**: Shape Classification. We generate a dataset of $32 \times 32$ grayscale images, where each image contains either a **circle** (label = 0) or a **square** (label = 1). The shapes are randomly positioned within the image, and each sample is annotated with a bounding box marking the region corresponding to the shape.

**Spurious Pixel Injection.** In the spurious test set, a bright pixel is introduced in the top-left corner of all images labeled $y = 1$, creating a distractor that may mislead the explanation.

**Model.** We train a CNN with two convolutional layers followed by fully connected layers. We also provide results of ResNets in Appendix L.

▷ **Synthetic Text Data**: Keyword-Based Classification. We construct a binary classification dataset where a sentence is labeled $y = 1$ if it contains the keywords "`apple`" or "`banana`", and $y = 0$ otherwise. This setup allows for a straightforward evaluation of whether explanations correctly highlight the relevant tokens. Note that the setting is slightly different from the illustration in Figure 1,3, which are slightly modified for better illustration.

**Spurious Token Injection.** In the spurious test set, a non-informative token (e.g., "`peach`") is inserted into sentences with $y = 1$ with 95% probability. This tests whether the model incorrectly attributes importance to the inserted token instead of the true predictive ones.

**Model.** We fine-tune a TinyBERT-based classifier (Jiao et al., 2020) that has two transformer layers.

## 5.2 MAIN RESULTS

Table 1 presents the results for quantified explanatory inversion based on the proposed IQ framework. Based on the results, we draw the following key insights.

**Explanatory inversion exists in all tasks, modalities, models, and explanation methods.** The baseline inversion scores (IS) are non-zero across all experiments, indicating that all explanation methods exhibit varying degrees of explanatory inversion, regardless of data modality (tabular, image, or text), model architecture (e.g., random forest, CNN, TinyBERT), or explanation method (e.g., SHAP, LIME, IG). The results also exhibit that explanatory inversion varies across models and tasks, which means it is hard to judge which post-hoc explanation method will suffer less from the inversion for a given model or task. However, we find that if the task is extremely simple, the inversion diminishes, as shown in Table 2 in Appendix H.

**Explanatory inversion is amplified by the injection of a spurious feature correlated with the output prediction.** The spurious feature injection during inference leads to a consistent increase in $R$, a decrease in $F$, and an overall increase in the inversion score (IS). This pattern is observed across

Table 1: Quantitative evaluation of explanation methods under normal and spurious scenarios across tabular, image, and text datasets. The scores (mean $\pm$ std) are averaged over 10 random seeds and reported as percentages (%). For each method, we measure $R$ (Reliance), $F$ (Faithfulness), IS (Inversion Score), and $A$ (Alignment) for both baseline and RBP. Additionally, results are provided for spurious test sets, where $\Delta$IS represents the change in inversion score between baseline and RBP. The results highlight that RBP consistently reduces the inversion score across all datasets and explanation methods, and indicate improved robustness against spurious features. See case studies in Appendix K.

| Model | Explanation | Baseline | | | | RBP | | | | Spurious Baseline | | | | Spurious RBP | | | | Baseline | RBP |
|---|---|---|---|---|---|---|---|---|---|---|---|---|---|---|---|---|---|---|---|
| | | $R(\downarrow)$ | $F(\uparrow)$ | IS$(\downarrow)$ | $A(\uparrow)$ | $R(\downarrow)$ | $F(\uparrow)$ | IS$(\downarrow)$ | $A(\uparrow)$ | $R(\downarrow)$ | $F(\uparrow)$ | IS$(\downarrow)$ | $A(\uparrow)$ | $R(\downarrow)$ | $F(\uparrow)$ | IS$(\downarrow)$ | $A(\uparrow)$ | $\Delta$IS$(\downarrow)$ | $\Delta$IS$(\downarrow)$ |
| **Tabular (Nonlinear Multi-Feature Regression)** | | | | | | | | | | | | | | | | | | | |
| Random Forest | SHAP | $8.6_{\pm0.2}$ | $67.1_{\pm0.3}$ | $24.2_{\pm0.1}$ | $88.5_{\pm0.2}$ | $7.8_{\pm0.1}$ | $81.5_{\pm0.2}$ | $13.9_{\pm0.2}$ | $92.0_{\pm0.1}$ | $11.3_{\pm0.2}$ | $47.1_{\pm0.3}$ | $38.0_{\pm0.2}$ | $83.1_{\pm0.2}$ | $9.6_{\pm0.1}$ | $60.3_{\pm0.2}$ | $28.5_{\pm0.2}$ | $87.6_{\pm0.3}$ | $14.0_{\pm0.1}$ | $14.4_{\pm0.2}$ |
| | LIME | $8.3_{\pm0.1}$ | $70.1_{\pm0.2}$ | $21.9_{\pm0.1}$ | $85.7_{\pm0.3}$ | $6.5_{\pm0.2}$ | $87.1_{\pm0.1}$ | $10.0_{\pm0.2}$ | $87.3_{\pm0.2}$ | $12.6_{\pm0.3}$ | $45.7_{\pm0.2}$ | $39.7_{\pm0.1}$ | $81.2_{\pm0.2}$ | $11.0_{\pm0.1}$ | $59.2_{\pm0.3}$ | $29.6_{\pm0.2}$ | $84.7_{\pm0.1}$ | $17.3_{\pm0.2}$ | $19.5_{\pm0.2}$ |
| SVM-RBF | SHAP | $7.4_{\pm0.2}$ | $67.0_{\pm0.3}$ | $23.9_{\pm0.1}$ | $88.0_{\pm0.2}$ | $7.0_{\pm0.1}$ | $78.1_{\pm0.2}$ | $16.0_{\pm0.2}$ | $90.2_{\pm0.1}$ | $11.7_{\pm0.2}$ | $46.2_{\pm0.3}$ | $38.6_{\pm0.2}$ | $84.5_{\pm0.1}$ | $10.1_{\pm0.1}$ | $59.8_{\pm0.2}$ | $29.0_{\pm0.2}$ | $86.1_{\pm0.3}$ | $14.9_{\pm0.1}$ | $12.8_{\pm0.2}$ |
| | LIME | $8.7_{\pm0.1}$ | $67.3_{\pm0.2}$ | $23.8_{\pm0.2}$ | $87.7_{\pm0.1}$ | $7.2_{\pm0.1}$ | $77.6_{\pm0.3}$ | $16.5_{\pm0.1}$ | $90.9_{\pm0.2}$ | $10.7_{\pm0.2}$ | $43.5_{\pm0.1}$ | $40.3_{\pm0.3}$ | $82.4_{\pm0.2}$ | $11.1_{\pm0.2}$ | $54.9_{\pm0.2}$ | $32.5_{\pm0.3}$ | $89.0_{\pm0.3}$ | $16.3_{\pm0.2}$ | $15.9_{\pm0.1}$ |
| MLP | SHAP | $8.6_{\pm0.2}$ | $64.6_{\pm0.3}$ | $25.9_{\pm0.1}$ | $88.2_{\pm0.2}$ | $6.4_{\pm0.1}$ | $74.5_{\pm0.2}$ | $18.3_{\pm0.2}$ | $92.6_{\pm0.1}$ | $12.5_{\pm0.2}$ | $41.3_{\pm0.3}$ | $42.1_{\pm0.2}$ | $83.1_{\pm0.2}$ | $10.9_{\pm0.1}$ | $55.1_{\pm0.2}$ | $32.4_{\pm0.2}$ | $90.3_{\pm0.1}$ | $16.4_{\pm0.1}$ | $13.9_{\pm0.2}$ |
| | LIME | $7.9_{\pm0.1}$ | $62.7_{\pm0.2}$ | $26.6_{\pm0.1}$ | $88.9_{\pm0.3}$ | $8.0_{\pm0.2}$ | $76.0_{\pm0.1}$ | $17.6_{\pm0.2}$ | $92.3_{\pm0.2}$ | $11.5_{\pm0.3}$ | $39.3_{\pm0.2}$ | $43.4_{\pm0.1}$ | $85.0_{\pm0.2}$ | $9.7_{\pm0.1}$ | $53.1_{\pm0.3}$ | $33.6_{\pm0.2}$ | $90.0_{\pm0.1}$ | $16.5_{\pm0.2}$ | $15.8_{\pm0.2}$ |
| | IG | $9.1_{\pm0.2}$ | $61.8_{\pm0.3}$ | $27.4_{\pm0.1}$ | $90.1_{\pm0.2}$ | $6.8_{\pm0.1}$ | $73.5_{\pm0.2}$ | $19.0_{\pm0.2}$ | $92.2_{\pm0.1}$ | $12.3_{\pm0.2}$ | $37.9_{\pm0.3}$ | $44.5_{\pm0.2}$ | $84.7_{\pm0.2}$ | $10.8_{\pm0.1}$ | $53.9_{\pm0.2}$ | $33.2_{\pm0.2}$ | $87.6_{\pm0.3}$ | $16.8_{\pm0.1}$ | $14.0_{\pm0.2}$ |
| | Occlusion | $11.0_{\pm0.1}$ | $64.0_{\pm0.2}$ | $26.3_{\pm0.2}$ | $91.1_{\pm0.3}$ | $6.5_{\pm0.2}$ | $74.1_{\pm0.1}$ | $18.6_{\pm0.2}$ | $93.5_{\pm0.2}$ | $11.9_{\pm0.3}$ | $40.5_{\pm0.2}$ | $42.6_{\pm0.1}$ | $86.3_{\pm0.2}$ | $10.9_{\pm0.1}$ | $53.7_{\pm0.3}$ | $33.3_{\pm0.2}$ | $90.7_{\pm0.1}$ | $16.1_{\pm0.2}$ | $14.5_{\pm0.2}$ |
| **Image (Shape Classification)** | | | | | | | | | | | | | | | | | | | |
| CNN | SHAP | $17.2_{\pm0.3}$ | $36.6_{\pm0.2}$ | $46.2_{\pm0.3}$ | – | $9.5_{\pm0.1}$ | $81.8_{\pm0.2}$ | $14.2_{\pm0.2}$ | – | $31.8_{\pm0.3}$ | $16.9_{\pm0.2}$ | $62.6_{\pm0.3}$ | – | $14.0_{\pm0.2}$ | $64.8_{\pm0.3}$ | $26.5_{\pm0.2}$ | – | $16.3_{\pm0.1}$ | $12.1_{\pm0.2}$ |
| | LIME | $19.3_{\pm0.2}$ | $26.3_{\pm0.3}$ | $53.6_{\pm0.2}$ | – | $15.6_{\pm0.2}$ | $35.4_{\pm0.1}$ | $46.7_{\pm0.3}$ | – | $35.3_{\pm0.2}$ | $19.5_{\pm0.2}$ | $61.9_{\pm0.3}$ | – | $17.2_{\pm0.3}$ | $21.4_{\pm0.2}$ | $56.6_{\pm0.2}$ | – | $8.1_{\pm0.1}$ | $9.7_{\pm0.2}$ |
| | IG | $11.1_{\pm0.2}$ | $38.6_{\pm0.3}$ | $43.8_{\pm0.2}$ | – | $7.7_{\pm0.1}$ | $43.6_{\pm0.2}$ | $39.9_{\pm0.1}$ | – | $19.1_{\pm0.2}$ | $25.4_{\pm0.3}$ | $54.2_{\pm0.2}$ | – | $11.4_{\pm0.1}$ | $29.6_{\pm0.2}$ | $50.1_{\pm0.3}$ | – | $10.1_{\pm0.2}$ | $10.0_{\pm0.1}$ |
| | Occlusion | $21.4_{\pm0.3}$ | $39.1_{\pm0.2}$ | $45.4_{\pm0.3}$ | – | $13.8_{\pm0.2}$ | $42.3_{\pm0.1}$ | $41.7_{\pm0.2}$ | – | $29.3_{\pm0.2}$ | $22.6_{\pm0.2}$ | $58.2_{\pm0.3}$ | – | $20.2_{\pm0.3}$ | $34.0_{\pm0.2}$ | $48.5_{\pm0.3}$ | – | $12.7_{\pm0.1}$ | $6.7_{\pm0.2}$ |
| **Text (Keyword-based Classification)** | | | | | | | | | | | | | | | | | | | |
| TinyBERT | SHAP | $7.3_{\pm0.1}$ | $14.0_{\pm0.2}$ | $60.7_{\pm0.3}$ | – | $1.0_{\pm0.1}$ | $15.8_{\pm0.2}$ | $59.2_{\pm0.2}$ | – | $45.8_{\pm0.3}$ | $10.8_{\pm0.2}$ | $70.6_{\pm0.3}$ | – | $31.8_{\pm0.2}$ | $18.8_{\pm0.3}$ | $61.4_{\pm0.2}$ | – | $9.8_{\pm0.1}$ | $2.1_{\pm0.1}$ |
| | LIME | $3.1_{\pm0.1}$ | $17.8_{\pm0.2}$ | $57.8_{\pm0.3}$ | – | $2.1_{\pm0.1}$ | $24.1_{\pm0.2}$ | $53.4_{\pm0.2}$ | – | $5.3_{\pm0.1}$ | $17.3_{\pm0.2}$ | $58.3_{\pm0.3}$ | – | $6.2_{\pm0.1}$ | $21.4_{\pm0.2}$ | $55.4_{\pm0.2}$ | – | $0.3_{\pm0.1}$ | $2.0_{\pm0.1}$ |
| | IG | $8.7_{\pm0.2}$ | $12.2_{\pm0.1}$ | $62.1_{\pm0.3}$ | – | $4.2_{\pm0.1}$ | $16.6_{\pm0.2}$ | $58.7_{\pm0.2}$ | – | $19.9_{\pm0.2}$ | $5.7_{\pm0.1}$ | $67.8_{\pm0.3}$ | – | $14.5_{\pm0.2}$ | $11.0_{\pm0.2}$ | $63.5_{\pm0.2}$ | – | $5.6_{\pm0.1}$ | $4.5_{\pm0.2}$ |

all models and explanation methods. For example, in tabular tasks, IS increases by more than $10\%$ under spurious conditions, validating that explanatory inversion intensifies.

**Increased inversion is associated with decreased alignment between attributions and ground-truth feature importance.** A clear negative correlation between IS and $A$ is observed. As IS increases under spurious conditions, $A$ consistently declines across models and explanation methods. This indicates that explanatory inversion not only reduces the reliability of the explanation but also worsens the alignment of attributions with true feature importance.

**RBP consistently mitigates explanatory inversion.** The application of RBP reduces the reliance score ($R$), increases faithfulness ($F$), and lowers the inversion score (IS). Moreover, RBP results in a smaller $\Delta$IS, indicating that explanation methods become more robust to spurious correlations. This is evidenced by the reduced $\Delta$IS values for RBP compared to the baseline across all modalities. Specifically, the average reduction in $\Delta$IS is approximately $0.95\%$ for tabular data, $2.19\%$ for image data, and $2.39\%$ for text data. This leads to a $1.8\%$ improvement across all the explanation methods and domains. Note that the improvement is consistent across domains and baselines. Importantly, the inversion score metric combines faithfulness and output reliance, and a $1.8\%$ gain translates to significant robustness under synthetic spurious attacks, as the absolute value of the inversion score remains relatively small (mostly smaller than 15% for tabular and image data and 5% for text data). These findings confirm that RBP enhances robustness against spurious correlations, particularly for tasks involving textual and image data.

## 5.3 IMPACT FROM VARIOUS SPURIOUS FEATURE INJECTION

To further analyze the properties of *explanatory inversion*, particularly focusing on spurious feature injection, we conduct a series of experiments to evaluate two key parameters in the IQ framework: (a) the number of spurious features injected and (b) the strength of the spurious feature ($\psi$). The experiment is conducted on the synthesized tabular data with MLP as the task model.

As shown in Figure 5 (a), increasing the number of spurious features leads to a consistent rise in $\Delta$IS across all explanation methods. This behavior aligns with the theoretical understanding that more spurious features can confuse the attribution process by creating multiple competing false explanations. Besides, we observe similar trends among different numbers of injected spurious features. By default, we set the number to 1 for simplicity and better illustration.

In Figure 5 (b), for a given spurious feature, we vary the spurious feature strength ($\psi$), which controls the degree of correlation between the injected spurious feature and the model's output. As $\psi$ increases, $\Delta$IS rises for all methods, indicating stronger explanatory inversion due to more pronounced spurious correlations. By default, we set the $\psi = 0.8$ for major experiments.

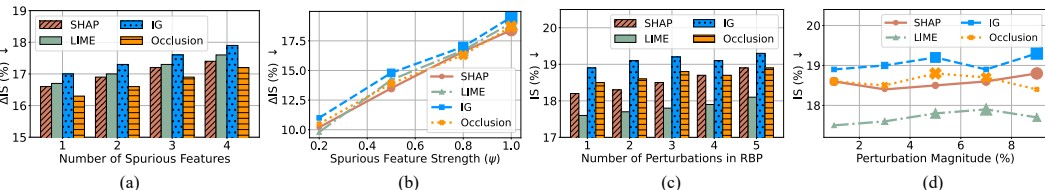

(a)        (b)        (c)        (d)

Figure 5: Ablation study analyzing (**i**). the properties of explanatory inversion via different spurious injection settings of IQ; and (**ii**) robustness and effectiveness of `RBP` under various hyper-parameter conditions. **(a)** Impact of varying the number of spurious features on $\Delta$IS. **(b)** Influence of spurious feature strength $\psi$ on $\Delta$IS. **(c)** Effect of the number of perturbation check in `RBP` on IS. **(d)** Influence of perturbation magnitude in `RBP` on IS.

## 5.4 PARAMETER SENSITIVITY ANALYSES OF RBP

To evaluate the sensitivity of `RBP` to different parameter choices, based on Section 4, on the tabular dataset with MLP model, we conduct experiments on two key parameters: (c) the number of perturbations applied in `RBP` and (d) the magnitude of perturbation noise. These parameters control the robustness and stability of `RBP`.

As shown in Figure 5 (c), increasing the number of perturbations in `RBP` has a relatively stable impact on IS across all explanation methods. While slight fluctuations are observed, particularly for the IG and SHAP methods, the overall stability suggests that `RBP` is robust to variations in the number of perturbations. Importantly, methods like Occlusion exhibit minor changes, indicating that the number of perturbations does not significantly degrade performance under these settings. In practice, we perform 3 perturbations for `RBP`.

In Figure 5 (d), we vary the perturbation noise magnitude from 1% to 9%. The results show that `RBP` maintains consistent IS values across this range, with only minor variations. This indicates that `RBP` is not overly sensitive to the exact magnitude of noise used for forward perturbations. The stability of IS across different noise levels highlights the adaptability of `RBP` in diverse scenarios. The results confirm that `RBP` balances perturbation-based refinement without introducing additional significant instability in attribution results. We use 5% magnitude for `RBP` by default.

## 5.5 GENERALIZATION TO REAL-WORLD APPLICATIONS

The proposed IQ framework for quantifying explanatory inversion is validated through a case study on a real-world dataset. Figure 6 illustrates the behavior of IG when applied to a ResNet-18 model on a CIFAR image correctly classified as a "dog". Under normal conditions, the attribution map highlights relevant areas of the image, including key features of the dog. However, when a spurious distractor pixel is injected, the explanation shifts toward this irrelevant feature, demonstrating an increased susceptibility

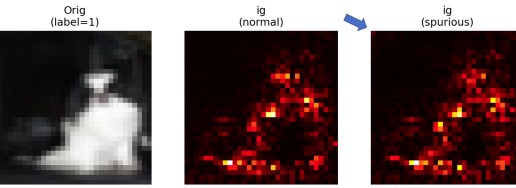

Figure 6: A case study of using IG for explaining a prediction from ResNet-18 (He et al., 2016) on the CIFAR Dataset. We observe that the injected pixel distractor is highlighted (you might need to enlarge the figure to see). See more results in Appendix L.

to explanatory inversion. This case study reveals that explanatory inversion is not limited to synthetic datasets or controlled scenarios but also manifests in real-world applications.

## 6 CONCLUSION

In this work, we introduce the *Inversion Quantification* (*IQ*) framework to measure *explanatory inversion*, where post-hoc explanations rely on model outputs rather than input-output relationships. Our analyses reveal that widely used methods like LIME and SHAP suffer from this issue across tabular, image, and text domains, especially under spurious correlations. To address this, we propose *Reproduce-by-Poking* (`RBP`), which integrates forward perturbation checks to reduce inversion. We show the effectiveness of `RBP` theoretically and empirically. Our findings emphasize the importance of reliable explanation methods in real-world applications. Future work will explore `RBP` for multimodal models and complex settings while maintaining scalability and computational efficiency.

ETHICS STATEMENT

We adhere to the ICLR Code of Ethics. No private, sensitive, or personally identifiable data is involved. Our work does not raise foreseeable ethical concerns or produce harmful societal outcomes.

REPRODUCIBILITY STATEMENT

Reproducibility is central to our work. All public datasets used in our experiments are standard benchmarks that are publicly available. We provide full details of the training setup, model architectures, and evaluation metrics in the main paper and appendix. Upon acceptance, we will release our codebase, including scripts for preprocessing, training, and evaluation, along with configuration files and documentation to facilitate exact reproduction of our results. Random seeds and hyperparameters will also be included to further ensure reproducibility.

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

CONTENTS OF THE APPENDIX

# A    PRELIMINARIES: POST-HOC EXPLANATION

In this section, we provide an overview of the explanation methods used across the tabular, image, and text domains, including SHAP, LIME, Integrated Gradients (IG), and Occlusion. Each method is formally defined as follows:

▷ **SHAP** (SHapley Additive exPlanations, Lundberg (2017)) is a method utilizing cooperative game theory that aims to fairly distribute the model's output among its input features based on their individual contributions. It attributes a model's output $f(x)$ for an input $x$ to features based on the Shapley value. The attribution for feature $i$ is defined as:

$$\phi_i = \sum_{S \subseteq \{1,...,d\} \setminus \{i\}} \frac{|S|!(d - |S| - 1)!}{d!} \left[ f(x_{S \cup \{i\}}) - f(x_S) \right], \tag{13}$$

where $x_S$ is the input restricted to the subset $S$ and $d$ is the total number of features. SHAP is versatile across tabular, image, and text data in providing feature-level attributions.

▷ **LIME** (Local Interpretable Model-Agnostic Explanations, Ribeiro et al. (2016)) explains a model's prediction by locally approximating it with a simpler interpretable model, typically a linear regression. For an input $x$, LIME constructs a surrogate model $g \in G$, where $g$ is a simple interpretable function (e.g., linear regression), and $G$ is the space of such interpretable models. The weights of $g$ correspond to the attributions of features. The explanation is generated by perturbing the input, evaluating the model $f$, and minimizing the following objective:

$$\arg \min_{g \in G} \sum_{x' \in \mathcal{N}(x)} \pi_x(x') \left( f(x') - g(x') \right)^2 + \Omega(g), \tag{14}$$

where $\mathcal{N}(x)$ is the set of perturbed samples, $\pi_x(x')$ is a proximity function measuring the similarity between $x'$ and $x$, and $\Omega(g)$ is a complexity penalty for the interpretable model $g$. LIME is model-agnostic and widely used for local explanations in multiple modalities, as it provides feature-level attributions based on the weights of the surrogate model.

▷ **IG** (Integrated Gradients, Sundararajan et al. (2017)) attributes the prediction difference between an input $x$ and a baseline $x'$ to each feature. The baseline $x'$ is a reference input that represents the absence of a meaningful signal, such as a black image for vision tasks or an empty/masked input for text. The attribution for feature $i$ is computed as:

$$\text{IG}_i = (x_i - x'_i) \int_{\alpha=0}^{1} \frac{\partial f(x' + \alpha(x - x'))}{\partial x_i} \, d\alpha. \tag{15}$$

IG accumulates gradients along a straight-line path from the baseline $x'$ to the actual input $x$, ensuring that attributions satisfy desirable properties like sensitivity and implementation invariance. This makes IG suitable for explaining high-dimensional models in image and text domains.

▷ **Occlusion** (Zeiler and Fergus (2014)) is a perturbation-based method that measures the change in the model's output when parts of the input are occluded. For a feature $i$, the attribution is:

$$\text{Occlusion}_i = f(x) - f(x^{-i}), \tag{16}$$

where $x^{-i}$ represents the input with feature $i$ occluded (e.g., replaced by a baseline value). Occlusion is particularly effective for visualizing feature importance in images and is applicable to tabular data.

These four methods were chosen because they are among the most widely used and are applicable across multiple models and domains. We acknowledge that other domain-specific methods, such as GradCAM (Selvaraju et al., 2017) for image data and attention scores for transformers, could also be employed for interpretation. However, we focus on general methods to maintain consistency across domains and leave the exploration of those domain-specific methods as future work. See Appendix B.

# B    DISCUSSION ON NEWER METHODS FOR POST-HOC EXPLANATIONS

Post-hoc explanation methods for machine learning models can be broadly categorized into several groups based on their underlying techniques and application domains. (1) Gradient-Based Explanation Methods: These methods leverage gradient information from deep learning models to highlight

influential features contributing to a prediction. Examples include Grad-CAM (Selvaraju et al., 2017), Score-CAM (Wang et al., 2020b), and Attention Flow in Transformers (Abnar and Zuidema, 2020). (2) Perturbation-Based Explanation Methods: These methods modify the input and analyze changes in the model's output to infer feature importance. Various works, including Rise (Petsiuk, 2018), S-LIME (Zhou et al., 2021), DLIME (Zafar and Khan, 2019), and DSEG-LIME (Knab et al., 2024), are proposed to improve stability, determinism, and segmentation in image-based explanations. (3) Counterfactual Explanations: These methods provide interpretability by generating alternative scenarios or feature importance scores. Examples include DiCE (Mothilal et al., 2020), CAL (Rao et al., 2021), TalkToModel (Slack et al., 2023), and DVCEs (Augustin et al., 2022). (4) Feature Attribution Methods: These methods propose to improve classic post-hoc feature attribution methods, including TreeSHAP (Lundberg and Lee, 2017), Kernel SHAP (Lundberg, 2017), Extended Kernel SHAP (Aas et al., 2021), along with the extension of SHAP to various scenarios (Fryer et al., 2021; Van den Broeck et al., 2022; Kumar et al., 2020).

In terms of explanations on natural language, recent advancements have proposed to explain language models in a post-hoc manner (Ding et al., 2022; Kaur et al., 2022; Kroeger et al., 2023; Krishna et al., 2024). Regarding the trustworthiness of post-hoc explanations, researchers have proposed to assess the effectiveness and fairness of explanation methods (Slack et al., 2021; Dai et al., 2022; Adebayo et al., 2022; Slack et al., 2020). More recently, researchers have considered incorporating human feedback to improve the quality of post-hoc explanations (Bianchi et al., 2024; Jesus et al., 2021; Han et al., 2022; Agarwal et al., 2021).

Another related field is causal explanation. Our spurious feature injection is motivated by causal considerations: the injected features are correlated with the output but non-causal. While our method does not require access to causal graphs or interventions, it aligns with the spirit of causal robustness. Causal explanation methods (Liu et al., 2023; 2024) rely on stronger assumptions (e.g., known or learnable structure), while our approach is diagnostic and model-agnostic. We will expand our related work section to better position our contributions in this context.

As a prosperous field, exhaustively experimenting with all post-hoc methods is impractical. In this paper, we choose the 4 most widely used methods, on which most of the later methods are built. Also, we only test on tabular, image, and text data. Comprehensive study on other modalities, including multi-modal scenarios, is a potential direction for future works.

## C CONTRAST TO PREVIOUS EVALUATION OF POST-HOC EXPLANATIONS

Evaluating the reliability and faithfulness of post-hoc explanations generated by methods like LIME, SHAP, and Integrated Gradients (IG) is a crucial area of XAI research. A significant body of work focuses on assessing whether explanations accurately reflect the model's internal reasoning process – often termed "faithfulness".

### C.1 ESTABLISHED FAITHFULNESS METRICS

Several metrics have been proposed to quantify faithfulness.

*Completeness*, as demonstrated by Integrated Gradients, ensures that attributions sum up to the difference between the model's output for the input and a baseline.

*Sensitivity* metrics assess how attributions change in response to input perturbations. For instance, sensitivity in IG relates to the axiom that if an input differs from the baseline in only one feature, and the model's prediction changes, that feature should receive non-zero attribution. Similarly, Yeh et al. (2019) define sensitivity concerning the explanation's response to small input perturbations.

*Perturbation-based* or *Deletion/Insertion tests* measure the change in model output when features deemed important (or unimportant) by the explanation are removed or added. Methods like Occlusion inherently operate on this principle, and benchmarks like those by Hooker et al. (2019) formalize this evaluation.

*Infidelity*, proposed by Yeh et al. (2019), measures the expected squared difference between the dot product of the input perturbation and the explanation, and the change in the model's output due to

that perturbation. It quantifies how well the explanation aligns with the model's local sensitivity to input changes.

## C.2  ROBUSTNESS AND STABILITY

Another line of evaluation focuses on the robustness or stability of explanations, examining how much attributions change under small, often imperceptible, perturbations to the input (Alvarez-Melis and Jaakkola, 2018). Work by Tan and Tian (2023) explores the potential trade-offs between explanation robustness and faithfulness. Jethani et al. (2023) shows that the explanation can be incorrectly related to the class label. While related to sensitivity, robustness specifically emphasizes the consistency and reliability of the explanation itself against minor input variations.

## C.3  EXPLANATORY INVERSION - A COMPLEMENTARY PERSPECTIVE

This paper introduces the concept of *Explanatory Inversion*, which describes a failure mode where explanations become overly conditioned on the model's output, potentially rationalizing predictions post-hoc rather than reflecting the forward reasoning process from inputs to outputs. We propose the *Inversion Quantification (IQ)* framework and the *Inversion Score $IS$* to measure this phenomenon. $IS$ combines two dimensions: *Reliance on Outputs $R$*, which quantifies the correlation between attributions and model predictions under perturbation, and *Explanation Faithfulness $F$*, which assesses alignment with the actual effect of features on the model's output using input perturbations.

Our proposed $IS$ metric offers a distinct perspective compared to existing faithfulness evaluations:

- $IS$ **vs. Infidelity/Sensitivity:** While infidelity and sensitivity focus on the alignment between explanation attributions and the effects of input perturbations on model output, $IS$ (particularly through the $R$ component) directly assesses the correlation between attributions and the model's output itself. Explanatory Inversion tackles the directionality of the explanation process – questioning if explanations are derived from the output rather than explaining how the output was reached. Therefore, $IS$ is conceptually different and not merely a special case of infidelity; it measures a distinct failure mode related to reversed justification.

- $IS$ **vs. Robustness/Stability:** Robustness measures the stability of attributions under input perturbations, whereas $IS$ measures the dependence of attributions on the output. While unstable explanations might exhibit higher inversion, the core focus differs; $IS$ specifically targets the potential for explanations to rationalize predictions.

- $IS$ **vs. Completeness/Deletion Tests:** Metrics like completeness and deletion tests verify if attributions account for the model's output difference or predict performance drops upon feature removal, respectively. They assess the forward impact assumption. $IS$, conversely, scrutinizes the potential reverse dependency – whether attributions are primarily dictated by the output, a scenario particularly relevant when spurious correlations exist.

## C.4  FORMAL COMPARISON OF RELIANCE ON OUTPUTS (R) AND INFIDELITY

This section provides a formal mathematical elaboration on the distinctions between the proposed Reliance on Outputs (R) metric, a component of the Inversion Score (IS), and the established metric of Infidelity. While both metrics utilize perturbations to evaluate explanations, they differ fundamentally in their mathematical formulation, the quantities they measure, and their conceptual goals in assessing explanation reliability.

### FORMAL DEFINITION OF INFIDELITY

Let $x \in \mathbb{R}^d$ be an input instance, $M : \mathbb{R}^d \to \mathbb{R}$ be the machine learning model producing an output $M(x)$, and $a = \mathcal{E}(x) \in \mathbb{R}^d$ be the attribution vector (explanation) for input $x$ generated by an explanation method $\mathcal{E}$. Infidelity measures how well the explanation $a$ captures the model's local behavior in response to perturbations.

Let $I \in \mathbb{R}^d$ be a perturbation vector drawn from a distribution $\mathcal{D}_I$ (e.g., Gaussian noise, or zeroing-out features). The infidelity of the explanation $a$ for input $x$ with respect to model $M$ and perturbation

distribution $\mathcal{D}_I$ is commonly defined as the expected squared difference between the change in model output predicted by the explanation and the actual change in model output:

$$\text{Infidelity}(M, \mathcal{E}, x, \mathcal{D}_I) = \mathbb{E}_{I \sim \mathcal{D}_I}\left[(a \cdot I - (M(x + I) - M(x)))^2\right]$$

Here, $a \cdot I$ represents the first-order approximation of the change in $M(x)$ based on the explanation $a$ if the input $x$ is changed by $I$. A low infidelity score suggests that the explanation $a$ is a locally faithful linear approximation of the model's behavior around $x$.

FORMAL DEFINITION OF RELIANCE ON OUTPUTS (R)

The Reliance on Outputs (R) metric is defined in the main paper (Definition 4.2) as a component of the Inversion Quantification (IQ) framework. It quantifies the degree to which attributions are influenced by the model's output rather than the input-output relationship, specifically by examining changes in attributions under feature-specific perturbations.

For a dataset $\mathcal{D} = \{(x_i, M(x_i))\}_{i=1}^N$, $R$ is defined as:

$$R = \frac{1}{N} \sum_{i=1}^N \frac{1}{d} \sum_{j=1}^d \rho(\Delta a_i^{(j)}, \Delta M(x_i; j))$$

where:

- $d$ is the number of features.
- $\rho$ is the correlation coefficient.
- $\Delta a_i^{(j)} = a_i^{(j)} - a_{base,i}^{(j)}$ represents the change in the attribution for feature $j$ of input $x_i$ after that feature $j$ is perturbed. Here, $a_i^{(j)}$ is the original attribution and $a_{base,i}^{(j)}$ is the attribution for feature $j$ after it has been perturbed in input $x_i$.
- $\Delta M(x_i; j) = M(x_i) - M(x_i^{(-j)})$ is the observed change in the model's output when feature $j$ is perturbed in input $x_i$ (where $x_i^{(-j)}$ is $x_i$ with feature $j$ perturbed).

A higher $R$ indicates a stronger linear association between changes in feature attributions and changes in model outputs when individual features are perturbed, suggesting a higher reliance of the explanation mechanism on the model's output.

FORMAL ELABORATION ON DIFFERENCES

Despite both leveraging perturbations, their mathematical foundations and objectives are distinct:

- **Nature of Perturbation:**
  - Infidelity: Typically employs random perturbations $I$ applied to the entire input space (e.g., $I \sim \mathcal{N}(0, \sigma^2 \mathbf{I})$ or feature masking/occlusion applied globally or in patches). These perturbations can affect multiple features simultaneously.
  - Reliance (R): Utilizes targeted, systematic perturbations applied to individual features one at a time ($x_j \rightarrow x_j'$), while other features $x_{k \neq j}$ are held constant.
- **Explanation Quantity Under Scrutiny:**
  - Infidelity: Evaluates the original, static attribution vector $a = \mathcal{E}(x)$ computed for the unperturbed input $x$. It tests the fidelity of this given $a$.
  - Reliance (R): Examines the change in the attribution value for the perturbed feature itself, i.e., $\Delta a^{(j)}$. It assesses how the explanation method $\mathcal{E}$ modifies its attribution for feature $j$ when that specific feature $j$ is altered in the input.
- **Mathematical Operation and Relationship Assessed:**
  - Infidelity: Computes $a \cdot I$, treating the explanation $a$ as defining a local linear model whose response to perturbation $I$ should approximate the model's response $M(x + I) - M(x)$. It measures the Mean Squared Error (MSE) of this approximation. The core operation is testing the predictive power of $a$ for $\Delta M$ under an input perturbation $I$.

- Reliance (R): Calculates the statistical correlation $\rho$ between two observed changes: the change in a specific feature's attribution ($\Delta a^{(j)}$) and the change in the model's output ($\Delta M(x; j)$), both arising from the same feature perturbation. It does not test $a^{(j)}$'s ability to predict $\Delta M(x; j)$ from the perturbation characteristics, but rather how $\Delta a^{(j)}$ covaries with $\Delta M(x; j)$.

ADDRESSING THE HYPOTHETICAL $R_{MSE}$ SCENARIO

If one were to define a variant of R using MSE instead of correlation, for instance:

$$R_{MSE} = \frac{1}{N} \sum_{i=1}^{N} \frac{1}{d} \sum_{j=1}^{d} \left( \lambda_1 \Delta a_i^{(j)} - \lambda_2 \Delta M(x_i; j) \right)^2$$

(where $\lambda_1, \lambda_2$ are potential scaling factors for commensurability, or simply 1 if units are compatible), this $R_{MSE}$ would measure the squared error between the (scaled) change in feature $j$'s attribution and the (scaled) change in model output when feature $j$ is perturbed. Even with this $R_{MSE}$ formulation, the fundamental distinctions from Infidelity persist:

- **Perturbation Type:** $R_{MSE}$ still relies on single-feature perturbations, whereas Infidelity typically uses global input perturbations.
- **Quantity Measured for Explanation:** $R_{MSE}$ still focuses on $\Delta a^{(j)}$ (the change/response of the attribution value itself), while Infidelity uses the original, full attribution vector $a$ to form $a \cdot I$.
- **Objective:** $R_{MSE}$ would assess if the explanation's change for a feature mirrors the output change due to perturbing that feature. Infidelity assesses if the original explanation vector correctly predicts output changes due to broader input perturbations.

CONCEPTUAL GOALS

- **Infidelity:** The primary goal of Infidelity is to quantify the local faithfulness or fidelity of an explanation $a$ to the model $M$. It answers: "Does the explanation $a$ accurately reflect how the model $M$ behaves in the local neighborhood of $x$?" Low infidelity indicates that the explanation is a good linear approximation of the model's function locally.
- **Reliance on Outputs (R):** The goal of R is to quantify a specific aspect of "explanatory inversion" – the degree to which the explanation generation process $\mathcal{E}$ appears to derive attributions by relying on the model's output $M(x)$ or changes thereof, rather than solely reflecting the model's learned input-to-output mapping. A high correlation in R suggests that the attribution for a feature changes primarily because perturbing that feature changes the model's output, which could be a sign that the explanation method is rationalizing the prediction from the output rather than explaining the decision process from the input.

## D  TIME COMPLEXITY

The time cost of RBP is approximately $k + 1$ times the cost of the base explainer, where $k$ is the number of perturbations. Our sensitivity analysis in Figure 5 (c) shows that a small value of $k$ (we use $k = 3$ by default) is sufficient to significantly reduce inversion and improve robustness across methods. We argue that this manageable, constant-factor increase in computation is a reasonable price for the significant gains in explanation reliability and robustness, especially in high-stakes applications where the cost of a misleading explanation can be far greater. We will clarify this trade-off more explicitly in the limitations section of the revised paper.

## E  PROOFS.

### E.1  PROOF OF THEOREM 3.5

**Restating Inversion Quantification.** Recall that explanatory inversion is defined as the degree to which attributions $\mathbf{a}$ depend on the model's output $M(\mathbf{x})$, rather than capturing the forward relationship between $\mathbf{x}$ and $M(\mathbf{x})$. To quantify this, the Inversion Score (IS) combines two components:

- **Reliance on Outputs** ($R$): Higher $R$ indicates stronger reliance of attributions on $M(\mathbf{x})$, reflecting potential backward explanations.

- **Explanation Faithfulness** ($F$): Higher $F$ reflects better alignment between attributions and the causal effect of features on $M(\mathbf{x})$.

The IS is defined as:

$$\text{IS}(R, F) = \left( \frac{R^p + (1 - F)^p}{2} \right)^{\frac{1}{p}},\tag{17}$$

where $p > 1$ is a hyperparameter controlling the sensitivity to deviations in $R$ and $F$.

**Proof of Dependency on** $\|\partial \mathbf{a}/\partial \mathbf{x}\|$. We aim to show that $\text{IS}(R, F) \propto \|\partial \mathbf{a}/\partial \mathbf{x}\|$.

*Proof.* The reliance score $R$ and faithfulness score $F$ are defined as follows:

$$R = \frac{1}{N} \sum_{i=1}^{N} \frac{1}{d} \sum_{j=1}^{d} \rho(\Delta a_i^{(j)}, \Delta M(\mathbf{x}_i; j)),\tag{18}$$

$$F = \frac{1}{N} \sum_{i=1}^{N} \frac{\sum_{j=1}^{d} a_i^{(j)} |\Delta M(\mathbf{x}_i; j)|}{\sum_{j=1}^{d} |a_i^{(j)}|}.\tag{19}$$

Step-by-step, we analyze how $R$ and $F$ depend on $\|\partial \mathbf{a}/\partial \mathbf{x}\|$:

**Step 1: Behavior of $R$.** The reliance score $R$ measures the correlation $\rho(\Delta a_i^{(j)}, \Delta M(\mathbf{x}_i; j))$, where $\Delta a_i^{(j)} = a_i^{(j)} - a_{\text{base}}^{(j)}$. The term $\Delta a_i^{(j)}$ is directly influenced by changes in $\mathbf{a}$ due to perturbations in $\mathbf{x}$. Specifically:

$$\Delta a_i^{(j)} \propto \frac{\partial \mathbf{a}}{\partial \mathbf{x}},\tag{20}$$

since small perturbations in $\mathbf{x}$ cause proportional changes in the attributions $\mathbf{a}$. Thus:

$$R \propto \|\partial \mathbf{a}/\partial \mathbf{x}\|.\tag{21}$$

**Step 2: Behavior of $F$.** The faithfulness score $F$ evaluates how well attributions align with the causal effect of features on $M(\mathbf{x})$. By definition:

$$F = \frac{1}{N} \sum_{i=1}^{N} \frac{\sum_{j=1}^{d} a_i^{(j)} |\Delta M(\mathbf{x}_i; j)|}{\sum_{j=1}^{d} |a_i^{(j)}|}.\tag{22}$$

Here:

$$\Delta M(\mathbf{x}_i; j) = M(\mathbf{x}_i) - M(\mathbf{x}_i^{(-j)}),\tag{23}$$

and $\Delta M(\mathbf{x}_i; j)$ is influenced by how sensitive the model's output is to changes in feature $j$. The attributions $a_i^{(j)}$, however, depend on $\mathbf{x}$ through:

$$a_i^{(j)} \propto \frac{\partial \mathbf{a}}{\partial \mathbf{x}}.\tag{24}$$

Hence, $F$ indirectly depends on $\|\partial \mathbf{a}/\partial \mathbf{x}\|$, as the quality of attributions is influenced by how changes in $\mathbf{x}$ align with the observed effects $\Delta M(\mathbf{x}_i; j)$.

**Step 3: Consequence for $\text{IS}(R, F)$.** From the definition

$$\text{IS}(R, F) = \left( \frac{R^p + (1 - F)^p}{2} \right)^{1/p},$$

we compute its partial derivatives:

$$\frac{\partial\, \text{IS}}{\partial R} = \frac{R^{p-1}}{2} \left( \frac{R^p + (1-F)^p}{2} \right)^{\frac{1}{p} - 1} \geq 0,$$

$$\frac{\partial\, \text{IS}}{\partial F} = -\frac{(1 - F)^{p-1}}{2} \left( \frac{R^p + (1-F)^p}{2} \right)^{\frac{1}{p} - 1} \leq 0.$$

Thus, IS is monotone nondecreasing in $R$ and monotone nonincreasing in $F$. From Steps 1 and 2, $R$ itself increases with attribution sensitivity $\|\partial \mathbf{a}/\partial \mathbf{x}\|$, while $F$ decreases with it. Since both of these monotone effects push IS upward, it follows that for sufficiently small perturbations, $\mathrm{IS}(R,F)$ increases whenever $\|\partial \mathbf{a}/\partial \mathbf{x}\|$ increases.

**Conclusion.** Therefore, the inversion score $\mathrm{IS}(R,F)$ grows monotonically with the attribution sensitivity $\|\partial \mathbf{a}/\partial \mathbf{x}\|$. In particular,

$$\mathrm{IS}(R,F) \;\propto\; \|\partial \mathbf{a}/\partial \mathbf{x}\|,$$

showing that explanatory inversion is amplified as attributions become more unstable with respect to the input. This completes the proof of Theorem 3.5. □

### E.2 Proof of Theorem 3.7

**Theorem 3.7.** *For any explanation method $\mathcal{E}$ and model $M$, if the ground-truth explanation $\mathbf{g}$ is available, then the alignment score $A$ upper-bounds explanation quality in terms of the Inversion Score* IS. *Specifically,*

$$A \;\leq\; 1 - \gamma \cdot \mathrm{IS}(R,F), \tag{25}$$

*for some constant $\gamma > 0$ that depends on the sensitivity of $A$ to attribution misalignment.*

*Proof.* We begin by normalizing the attribution and ground-truth vectors so that $\|\mathbf{a}\| = \|\mathbf{g}\| = 1$. This ensures that the alignment score reduces to the simple inner product $A = \langle \mathbf{a}, \mathbf{g} \rangle$. Writing the attribution error vector as $\boldsymbol{\epsilon} = \mathbf{a} - \mathbf{g}$, we expand

$$\|\boldsymbol{\epsilon}\|^2 = \|\mathbf{a}\|^2 + \|\mathbf{g}\|^2 - 2\langle \mathbf{a}, \mathbf{g} \rangle = 2(1 - A),$$

so that

$$A = 1 - \tfrac{1}{2}\|\boldsymbol{\epsilon}\|^2. \tag{26}$$

Thus, alignment decreases quadratically in the magnitude of the attribution error.

Next, we connect the Inversion Score to this error. We conceptually decompose $\boldsymbol{\epsilon}$ into two orthogonal components, one attributable to reliance on outputs ($P_R \boldsymbol{\epsilon}$) and the other to lack of faithfulness ($P_F \boldsymbol{\epsilon}$). By construction of this decomposition, the two components together control the full error in the sense that

$$\|P_R \boldsymbol{\epsilon}\| + \|P_F \boldsymbol{\epsilon}\| \;\geq\; \mu \|\boldsymbol{\epsilon}\|,$$

for some $\mu \in (0,1]$ measuring how tightly the projections approximate the total error.

Now, by calibration of the reliance and faithfulness metrics, small attribution errors in the $P_R$ and $P_F$ directions translate proportionally into changes in $R$ and $F$. That is, there exist constants $k_R, k_F > 0$ such that

$$R \;\geq\; k_R \|P_R \boldsymbol{\epsilon}\|, \qquad 1 - F \;\geq\; k_F \|P_F \boldsymbol{\epsilon}\|,$$

whenever $\|\boldsymbol{\epsilon}\|$ is small enough. Substituting these inequalities into the definition of the Inversion Score,

$$\mathrm{IS}(R,F) = \left( \tfrac{R^p + (1-F)^p}{2} \right)^{1/p}, \quad p > 1,$$

yields the lower bound

$$\mathrm{IS}(R,F) \;\geq\; \left( \tfrac{(k_R \|P_R \boldsymbol{\epsilon}\|)^p + (k_F \|P_F \boldsymbol{\epsilon}\|)^p}{2} \right)^{1/p}.$$

Applying the standard inequality between $\ell_p$ and $\ell_1$ norms gives

$$\mathrm{IS}(R,F) \;\geq\; \frac{k_R \|P_R \boldsymbol{\epsilon}\| + k_F \|P_F \boldsymbol{\epsilon}\|}{2^{\,1-1/p}}.$$

Combining this with the earlier decomposition property, we conclude

$$\mathrm{IS}(R,F) \;\geq\; c_{\mathrm{IS}} \|\boldsymbol{\epsilon}\|, \qquad c_{\mathrm{IS}} := \frac{\mu \min\{k_R, k_F\}}{2^{\,1-1/p}} > 0. \tag{27}$$

This inequality shows that a larger inversion score necessarily implies a larger attribution error.

Finally, substituting equation 27 into the alignment formula equation 26, we obtain

$$A = 1 - \tfrac{1}{2}\|\epsilon\|^2 \ \leq \ 1 - \frac{1}{2c_{\mathrm{IS}}^2}\,\mathrm{IS}(R,F)^2.$$

Denoting $\gamma_2 = \frac{1}{2c_{\mathrm{IS}}^2}$, this gives the quadratic bound

$$A \ \leq \ 1 - \gamma_2\,\mathrm{IS}(R,F)^2.$$

Since $\mathrm{IS}(R,F) \in [0,1]$, the quadratic term can be relaxed to a linear one. For any $\lambda \in (0,1]$ and $\mathrm{IS} \geq \lambda$, the inequality $\mathrm{IS}^2 \geq \lambda \cdot \mathrm{IS}$ holds, yielding

$$A \ \leq \ 1 - (\gamma_2\lambda)\,\mathrm{IS}(R,F).$$

Setting $\gamma := \gamma_2\lambda > 0$ recovers the claimed linear upper bound.

**Conclusion.** We have shown that alignment is bounded above by a decreasing function of the inversion score, with constants depending on the decomposition tightness $\mu$, calibration factors $k_R, k_F$, and the power $p$. In particular,

$$A \ \leq \ 1 - \gamma \cdot \mathrm{IS}(R,F),$$

which establishes Theorem 3.7. $\qquad\qquad\square$

### E.3 Theoretical Justification for Theorem 3.7

To justify the upper bound between explanation alignment $A$ and Inversion Score $\mathrm{IS}(R,F)$ in Theorem 3.7, we turn to recent insights from information theory and algorithmic complexity (Rao, 2025). Specifically, we show that explanations that suffer from high explanatory inversion necessarily incur inefficiencies in information encoding and fidelity, which manifests as a measurable drop in alignment with ground-truth causal attributions.

**Kolmogorov Complexity and Information Inefficiency.** Let $g$ denote the explanation function used to produce the attribution vector $\mathbf{a} = g(x)$. Let $K(g)$ denote the Kolmogorov complexity of this explanation function, and $I(X; g(X))$ be the mutual information between the input $X$ and the generated explanation $g(X)$. From prior work in algorithmic information theory, it is known that the expected explanation error obeys the following lower bound:

$$\mathbb{E}_{\mathcal{D}}[d(f(X), g(X))] \geq \Omega\left(2^{-(K(g)-I(X;g(X)))}\right), \tag{28}$$

indicating that when $g$ fails to efficiently use its information budget (i.e., when $K(g) \gg I(X; g(X))$), the explanation error increases exponentially.

In our context, this inefficiency translates to a higher attribution error $\|\mathbf{a} - \mathbf{g}\|$, and consequently, a lower alignment score $A = \langle \mathbf{a}, \mathbf{g} \rangle = 1 - \tfrac{1}{2}\|\mathbf{a} - \mathbf{g}\|^2$.

**Bounding Misalignment via Complexity Gap.** Furthermore, prior results show that if $K(g) < K(f) - c$ for a model $f$, then there exists at least one input $x$ such that $f(x) \neq g(x)$ with high probability. This result implies that an explanation $g$ with significantly lower complexity than the model it seeks to approximate cannot achieve high fidelity. Thus, if $K(g)$ is significantly smaller than $K(f)$, then alignment must degrade:

$$1 - A \geq \lambda \cdot (K(f) - K(g)), \tag{29}$$

for some constant $\lambda > 0$ depending on the smoothness of $f$ and the explanation resolution.

**Relation to Inversion Score.** Recall that the Inversion Score is defined as:

$$\mathrm{IS}(R,F) = \left(\frac{R^p + (1-F)^p}{2}\right)^{1/p}, \tag{30}$$

where high $R$ and low $F$ reflect greater reliance on model outputs and weaker causal faithfulness, respectively. We interpret $\mathrm{IS}(R, F)$ as a proxy for the extent to which the explanation has diverged from input-grounded causal reasoning toward model-output-driven heuristics.

Such divergence implies that $g(X)$ has reduced mutual information with $X$, i.e., $I(X; g(X))$ is small, while the representation $g$ may still have large complexity $K(g)$ if it encodes spurious correlations. Consequently, the complexity gap $K(g) - I(X; g(X))$ becomes large, increasing the expected attribution error and hence lowering $A$.

Putting this together:

$$A = 1 - \frac{1}{2}\|\mathbf{a} - \mathbf{g}\|^2 \leq 1 - \gamma \cdot \mathrm{IS}(R, F)^2, \tag{31}$$

for some $\gamma > 0$ that incorporates the relationship between complexity inefficiency and explanatory inversion.

**Connection to Rate-Distortion Theory.** Finally, from the perspective of lossy encoding, explanation fidelity can be viewed through the lens of rate-distortion theory. Let $R_f(\delta)$ denote the rate-distortion function of model $f$ for a target distortion $\delta$. Then, an explanation $g$ achieving distortion at most $\delta$ must satisfy:

$$K(g) \geq R_f(\delta) - O(1). \tag{32}$$

This implies that if an explanation's complexity $K(g)$ is insufficient, the distortion must increase. Since distortion directly increases attribution error and thus decreases $A$, we again see that misalignment is penalized.

Thus, the upper bound between alignment $A$ and Inversion Score $\mathrm{IS}(R, F)$ in Theorem 3.7 is not only intuitive, but also consistent with the information-theoretic limits of explainability. It reflects the fundamental cost of deviating from causal fidelity and grounding in input semantics.

### E.4 PROOFS OF THEORETICAL PROPERTIES OF RBP

In this subsection, we provide formal proofs for Theorems 4.1(a), 4.1(b), and 4.2. These theorems establish key guarantees of our proposed RBP: reduced reliance on the model's outputs, improved attribution faithfulness, and resilience to spurious features. Before we present the detailed proofs, we first formulate the setup and a lemma.

**Formal Setup.** Let $M : \mathbb{R}^d \to \mathbb{R}$ be a fixed model, $E$ an attribution method, and $a = E(x) \in \mathbb{R}^d$ the attribution for input $x$. For feature $j$:

- $\Delta M(x; j)$ denotes the measured change in $M(x)$ when feature $j$ is perturbed (evaluation perturbation).
- $\Delta a^{(j)}$ is the corresponding change in attribution $a^{(j)}$.
- The reliance score aggregates correlations $\mathrm{Corr}(\Delta a^{(j)}, \Delta M(x; j))$ across features and samples.
- The faithfulness score is the attribution-weighted output sensitivity:

$$F = \frac{\sum_j |a^{(j)}| |\Delta M(x; j)|}{\sum_j |a^{(j)}|}. \tag{33}$$

- For RBP, we estimate a deviation score $\delta^{(j)} \geq 0$ from prediction-preserving perturbations and define refined attributions

$$\tilde{a}^{(j)} = \frac{a^{(j)}}{1 + \lambda \delta^{(j)}}, \quad \lambda > 0. \tag{34}$$

To begin with, we introduce the following lemma that will be used for the proof of Theorem 4.1(b) and 4.2.

**Lemma E.1** (Causal–spurious separation). *Let $M : \mathbb{R}^d \to \mathbb{R}$ be twice differentiable in a neighborhood of $x$ with gradient $\mathbf{g} = \nabla M(x) \neq 0$ and Hessian bounded componentwise by $\|H(\xi)\|_\infty \leq L_H$ for all $\xi$ in that neighborhood. Consider prediction-preserving perturbations drawn uniformly from*

$$\mathcal{S}_\varepsilon := \{\Delta x \in \mathbb{R}^d : \|\Delta x\| \leq \varepsilon, \ \mathbf{g}^\top \Delta x = 0\},$$

*and evaluation perturbations that change only coordinate $j$ by $\eta > 0$. Let $E$ be an attribution method that is locally coordinate-wise Lipschitz at $x$: for each $j$ there exists $L_j > 0$ such that $|a^{(j)}(x + \Delta x) - a^{(j)}(x)| \leq L_j |\Delta x_j|$ for all $\|\Delta x\|$ small. Define*

$$\delta^{(j)} := \mathbb{E}_{\Delta x \sim \mathrm{Unif}(\mathcal{S}_\varepsilon)} \big| a^{(j)}(x + \Delta x) - a^{(j)}(x) \big|.$$

*Then for all sufficiently small $\varepsilon, \eta$, the following hold with explicit constants:*

$$\textbf{(Spurious } j\textbf{)} \quad g_j = 0 \implies \delta^{(j)} \geq c_{\mathrm{spur}} \, \varepsilon, \qquad |\Delta M(x; j)| \leq C_2 \, \eta^2, \tag{35}$$

$$\textbf{(Causal } j\textbf{)} \quad g_j \neq 0 \implies \delta^{(j)} \leq c_{\mathrm{caus}} \, \varepsilon, \qquad |\Delta M(x; j)| \geq C_1 \, \eta, \tag{36}$$

*where*

$$c_{\mathrm{spur}} := L_j \, \frac{(d + 3)}{12 \sqrt{2} \, (d + 1)^{3/2}}, \qquad c_{\mathrm{caus}} := L_j \, \alpha_j, \; \alpha_j := \sqrt{\frac{\|\mathbf{g}\|^2 - g_j^2}{(d + 1) \, g_j^2}},$$

$$C_2 := \tfrac{1}{2} L_H, \qquad C_1 := \tfrac{1}{2} |g_j|.$$

*In particular, for small perturbations, spurious features have near-zero output effect and larger deviation scores, whereas causal features have output effect bounded away from zero and smaller deviation scores.*

*Proof.* **Step 1: Geometry and isotropy on $\mathcal{S}_\varepsilon$.** Let $e_1 = \mathbf{g} / \|\mathbf{g}\|$ and extend to an orthonormal basis $\{e_1, \ldots, e_d\}$. Write $\Delta x = \sum_{i=1}^d \Delta x_i e_i$. Then $\mathbf{g}^\top \Delta x = \|\mathbf{g}\| \, \Delta x_1$, so the constraint $\mathbf{g}^\top \Delta x = 0$ is equivalent to $\Delta x_1 = 0$. Hence

$$\mathcal{S}_\varepsilon = \Big\{ (\Delta x_1, \ldots, \Delta x_d) : \; \Delta x_1 = 0, \; \sum_{i=2}^d \Delta x_i^2 \leq \varepsilon^2 \Big\},$$

i.e., a solid $(d - 1)$-ball in the subspace orthogonal to $\mathbf{g}$. Let $m := d - 1$ and denote by $U = (U_2, \ldots, U_d)$ a random vector uniformly distributed in the $m$-ball of radius $\varepsilon$. By rotational symmetry,

$$\mathbb{E}[U_i] = 0, \qquad \mathrm{Var}(U_i) = \frac{\varepsilon^2}{m + 2} = \frac{\varepsilon^2}{d + 1} \quad \text{for each } i \in \{2, \ldots, d\}. \tag{37}$$

Moreover, the fourth moment is (standard integral for the uniform ball)

$$\mathbb{E}[U_i^4] = \frac{3 \, \varepsilon^4}{(m + 2)(m + 4)} = \frac{3 \, \varepsilon^4}{(d + 1)(d + 3)}. \tag{38}$$

**Step 2: A uniform lower bound on $\mathbb{E}|U_i|$.** Apply Paley–Zygmund to the nonnegative variable $Z := U_i^2$ with parameter $\theta := \tfrac{1}{2}$:

$$\mathbb{P}\big( U_i^2 \geq \tfrac{1}{2} \mathbb{E}[U_i^2] \big) \geq \frac{(1 - \theta)^2 \, \mathbb{E}[U_i^2]^2}{\mathbb{E}[U_i^4]} = \frac{\tfrac{1}{4} \cdot (\varepsilon^4 / (d + 1)^2)}{3 \, \varepsilon^4 / ((d + 1)(d + 3))} = \frac{d + 3}{12(d + 1)}.$$

Therefore,

$$\mathbb{E}|U_i| \geq \sqrt{\tfrac{1}{2} \mathbb{E}[U_i^2]} \cdot \mathbb{P}\big( U_i^2 \geq \tfrac{1}{2} \mathbb{E}[U_i^2] \big) \geq \frac{d + 3}{12 \sqrt{2} \, (d + 1)^{3/2}} \, \varepsilon. \tag{39}$$

**Step 3: Spurious coordinate ($g_j = 0$).** If $g_j = 0$, then the $j$–axis lies entirely in the free subspace $\mathrm{span}\{e_2, \ldots, e_d\}$, so $\Delta x_j$ is distributed as $U_i$ for some $i \in \{2, \ldots, d\}$. By the Lipschitz property of $E$ and equation 39,

$$\delta^{(j)} = \mathbb{E}|a^{(j)}(x + \Delta x) - a^{(j)}(x)| \geq L_j \, \mathbb{E}|\Delta x_j| \geq L_j \, \frac{d + 3}{12 \sqrt{2} \, (d + 1)^{3/2}} \, \varepsilon =: c_{\mathrm{spur}} \, \varepsilon,$$

proving the spurious deviation bound in equation 35.

For the evaluation perturbation along $e_j$, a second-order Taylor expansion gives

$$\Delta M(x; j) = M(x + \eta e_j) - M(x) = g_j \, \eta + \tfrac{1}{2} \, H_{jj}(\xi) \, \eta^2$$

for some $\xi$ on the segment $[x, x + \eta e_j]$. If $g_j = 0$, the linear term vanishes and so

$$|\Delta M(x; j)| \leq \tfrac{1}{2} |H_{jj}(\xi)| \eta^2 \leq \tfrac{1}{2} L_H \eta^2 =: C_2 \eta^2,$$

which is the second part of equation 35.

**Step 4: Causal coordinate ($g_j \neq 0$).** When $g_j \neq 0$, the hyperplane constraint couples $\Delta x_j$ to the remaining free coordinates:

$$g_j \Delta x_j + \sum_{i \neq 1, j} g_i \Delta x_i = 0 \implies \Delta x_j = -\frac{1}{g_j} \sum_{i \neq 1, j} g_i \Delta x_i.$$

Using symmetry (zero cross-covariances) and equation 37,

$$\mathrm{Var}(\Delta x_j) = \frac{1}{g_j^2} \sum_{i \neq 1, j} g_i^2 \, \mathrm{Var}(\Delta x_i) = \frac{\|\mathbf{g}\|^2 - g_j^2}{g_j^2} \cdot \frac{\varepsilon^2}{d + 1}.$$

Hence, by Cauchy–Schwarz,

$$\mathbb{E}|\Delta x_j| \leq \sqrt{\mathrm{Var}(\Delta x_j)} = \sqrt{\frac{\|\mathbf{g}\|^2 - g_j^2}{(d + 1) g_j^2}} \, \varepsilon =: \alpha_j \, \varepsilon.$$

Applying Lipschitzness of $E$ yields

$$\delta^{(j)} \leq L_j \, \mathbb{E}|\Delta x_j| \leq L_j \, \alpha_j \, \varepsilon =: c_{\mathrm{caus}} \varepsilon,$$

which is the causal deviation bound in equation 36.

For the evaluation perturbation, the same Taylor formula gives

$$\Delta M(x; j) = g_j \, \eta + \tfrac{1}{2} \, H_{jj}(\xi) \, \eta^2.$$

Choose $\eta_0 := |g_j| / L_H$. For all $0 < \eta \leq \eta_0$,

$$|\Delta M(x; j)| \geq |g_j| \, \eta - \tfrac{1}{2} L_H \, \eta^2 \geq \tfrac{1}{2} |g_j| \, \eta =: C_1 \, \eta,$$

establishing the output lower bound in equation 36.

**Step 5: Separation and conclusion.** Summarizing, for sufficiently small $\varepsilon, \eta$:

$$\begin{cases} g_j = 0: & \delta^{(j)} \geq c_{\mathrm{spur}} \varepsilon, & |\Delta M(x; j)| \leq C_2 \eta^2, \\ g_j \neq 0: & \delta^{(j)} \leq c_{\mathrm{caus}} \varepsilon, & |\Delta M(x; j)| \geq C_1 \eta. \end{cases}$$

Thus, spurious features have (i) near-zero output sensitivity and (ii) larger deviation scores, while causal features have (i) output sensitivity bounded away from zero and (ii) smaller deviation scores. $\square$

**Assumptions.** Additionally, we make the following assumptions for our proofs:

**A1 (Prediction-preserving perturbations).** The perturbations used to compute $\delta^{(j)}$ satisfy $|M(x) - M(x')| \leq \eta$ for tolerance $\eta \geq 0$. Randomness of these perturbations is independent (to first order) of the evaluation perturbations that yield $\Delta M(x; j)$.

**A2 (Deviation monotonicity).** $\delta^{(j)}$ is a nonnegative measure of attribution instability: if $\mathrm{Var}(E(x)^{(j)})$ under prediction-preserving perturbations increases, then $\delta^{(j)}$ increases.

With the above lemma and assumptions, now we present the detailed proofs of Theorem 4.1(a) and Theorem 4.1(b).

**Theorem 4.1(a).** *RBP reduces reliance on outputs: there exists an explicit $\lambda_0 > 0$ such that for all $\lambda \in (0, \lambda_0]$, $R_{\mathrm{RBP}} < R$.*

*Proof.* **Setup.** Fix a feature $j$ and write

$$X = \Delta a^{(j)}, \qquad Y = \Delta M(x; j), \qquad Z = \delta^{(j)} \, (\geq 0).$$

The refined correlation for feature $j$ under shrinkage parameter $\lambda$ is

$$\phi_j(\lambda) \;=\; \mathrm{Corr}\Big(\frac{X}{1 + \lambda Z}, \, Y\Big).$$

At $\lambda = 0$, this reduces to the baseline correlation $\phi_j(0) = \rho(X, Y)$.

**Derivative at $\lambda = 0$.** A standard correlation-derivative calculation gives

$$\phi_j'(0) = -\frac{\mathrm{Cov}(XZ, Y)}{\sigma_X \sigma_Y} + \frac{1}{2} \, \rho(X, Y) \, \frac{\mathrm{Var}(XZ)}{\sigma_X^2}, \tag{40}$$

where $\sigma_X^2 = \mathrm{Var}(X)$ and $\sigma_Y^2 = \mathrm{Var}(Y)$.

By Assumption A1, $Z$ is independent (to first order) of $Y$, and by Assumption A2, $Z$ is larger when $X$ is unstable. Together, these ensure that the leading covariance term is strictly negative. Thus there exists $c_j > 0$ such that

$$\phi_j'(0) \;\leq\; -c_j \;<\; 0.$$

**Bounding higher-order terms.** To move beyond $\lambda = 0$, we use Taylor's theorem with remainder:

$$\phi_j(\lambda) = \phi_j(0) + \lambda \phi_j'(0) + \tfrac{1}{2}\lambda^2 \phi_j''(\xi) \quad \text{for some } \xi \in (0, \lambda).$$

To control the remainder, assume finite moments: there exist constants $\Delta_{\max}, C_2, C_4 > 0$ such that $0 \leq Z \leq \Delta_{\max}$, $\mathbb{E}[X^2 Z^2] \leq C_2$, $\mathbb{E}[X^4 Z^4] \leq C_4$, and $\sigma_X, \sigma_Y$ are bounded away from 0. Then a direct calculation (chain and quotient rules, with Cauchy–Schwarz) yields a uniform bound

$$|\phi_j''(\lambda)| \;\leq\; K_j \quad \text{for all } \lambda \in [0, 1/\Delta_{\max}], \tag{41}$$

where $K_j$ is finite and can be expressed in terms of $C_2, C_4, \sigma_X, \sigma_Y$, and $|\rho|$.

**Explicit range for $\lambda$.** Combining the expansion and the bounds gives

$$\phi_j(\lambda) - \phi_j(0) \;\leq\; -c_j \lambda + \tfrac{1}{2} K_j \lambda^2.$$

Hence $\phi_j(\lambda) < \phi_j(0)$ whenever

$$0 < \lambda < \frac{2c_j}{K_j}.$$

Since denominators $1 + \lambda Z$ remain positive provided $\lambda \leq 1/\Delta_{\max}$, we define

$$\lambda_{0,j} = \min\Big\{ \tfrac{2c_j}{K_j}, \tfrac{1}{\Delta_{\max}} \Big\}.$$

Thus for every $\lambda \in (0, \lambda_{0,j})$, the refined correlation for feature $j$ is strictly smaller: $\phi_j(\lambda) < \phi_j(0)$.

**From individual features to global reliance.** The reliance scores $R$ and $R_{\mathrm{RBP}}$ are defined as averages of these correlations across features and samples. Therefore, setting

$$\lambda_0 = \min_j \lambda_{0,j},$$

we conclude that for all $\lambda \in (0, \lambda_0)$,

$$R_{\mathrm{RBP}} < R.$$

$\square$

**Theorem 4.1(b).** *RBP improves faithfulness:* $F_{\mathrm{RBP}} \geq F$.

*Proof.* **Setup.** The refined faithfulness score under RBP is

$$F_{\mathrm{RBP}} \;=\; \frac{\sum_j \alpha_j w_j u_j}{\sum_j \alpha_j w_j},$$

where $\alpha_j = |a^{(j)}| \geq 0$ is the attribution magnitude, $u_j = |\Delta M(x;j)| \geq 0$ is the model's measured sensitivity to feature $j$, and $w_j = (1 + \lambda\delta^{(j)})^{-1} \in (0,1]$ is the stability weight, with $\delta^{(j)} \geq 0$. The baseline score is

$$F = \frac{\sum_j \alpha_j u_j}{\sum_j \alpha_j}.$$

We assume throughout that the denominators $\sum_j \alpha_j$ and $\sum_j \alpha_j w_j$ are strictly positive (non-degeneracy).

**Reduction to an inequality.** To compare $F_{\mathrm{RBP}}$ and $F$, consider

$$D = \Big(\sum_j \alpha_j w_j u_j\Big)\Big(\sum_j \alpha_j\Big) - \Big(\sum_j \alpha_j u_j\Big)\Big(\sum_j \alpha_j w_j\Big).$$

Note that $F_{\mathrm{RBP}} \geq F$ if and only if $D \geq 0$.

**Symmetrization.** Expanding $D$ and rearranging gives

$$D = \frac{1}{2}\sum_{i,j} \alpha_i \alpha_j \Big[(w_i - w_j)(u_i - u_j)\Big].$$

This identity is exact: it follows by multiplying out both sides and symmetrizing terms.

**Sign of each summand.** By construction $\alpha_i \alpha_j \geq 0$. Thus the sign of each term is governed by $(w_i - w_j)(u_i - u_j)$. Lemma E.1 states that more causal features (larger $u_j$) tend to have smaller deviation $\delta^{(j)}$, and hence larger weight $w_j$. That is, the mapping $j \mapsto (u_j, w_j)$ is positively associated: whenever $u_i > u_j$, we also have $w_i \geq w_j$. Consequently,

$$(w_i - w_j)(u_i - u_j) \geq 0 \quad \text{for all } i,j.$$

**Conclusion of inequality.** Since each summand in the symmetrized sum is nonnegative, we obtain $D \geq 0$. Therefore, $F_{\mathrm{RBP}} \geq F$. $\qquad\square$

**Theorem 4.2.** *RBP down-weights spurious features: for any spurious feature $j$, if $\delta^{(j)} \geq \Delta_{\min}$ or $|\Delta M(x;j)| \leq \epsilon$, then the refined attribution satisfies $\tilde{a}^{(j)} \leq \frac{a^{(j)}}{1 + \lambda\Delta_{\min}}$ and contributes negligibly to $F_{\mathrm{RBP}}$.*

*Proof.* **Setup.** For a feature $j$, the baseline attribution is $a^{(j)}$, and the refined attribution under RBP is

$$\tilde{a}^{(j)} = \frac{a^{(j)}}{1 + \lambda\delta^{(j)}}, \qquad \lambda > 0,$$

where $\delta^{(j)} \geq 0$ measures attribution instability. The contribution of this feature to the refined faithfulness score is

$$C_j = \frac{|\tilde{a}^{(j)}|}{\sum_k |\tilde{a}^{(k)}|}\,|\Delta M(x;j)|.$$

We analyze two cases, corresponding to the two conditions in the theorem.

**Case 1: Large deviation.** Suppose feature $j$ is spurious and highly unstable under prediction-preserving perturbations. By Assumption A2, this instability implies

$$\delta^{(j)} \geq \Delta_{\min} > 0.$$

Substituting into the refinement rule gives

$$\tilde{a}^{(j)} = \frac{a^{(j)}}{1 + \lambda\delta^{(j)}} \leq \frac{a^{(j)}}{1 + \lambda\Delta_{\min}}.$$

Thus, the refined attribution is upper-bounded by a shrinkage factor depending only on $\lambda$ and the stability threshold $\Delta_{\min}$. As $\Delta_{\min} \to \infty$, this bound goes to zero:

$$\lim_{\Delta_{\min}\to\infty} \tilde{a}^{(j)} = 0.$$

Hence, spurious features with high deviation are driven arbitrarily close to zero weight.

**Case 2: Small output effect.** Now suppose feature $j$ has negligible causal influence on the output. By Lemma E.1, this means

$$|\Delta M(x; j)| \leq \epsilon,$$

for some small $\epsilon > 0$. Then its contribution to the refined faithfulness score satisfies

$$C_j \ \leq \ \frac{|a^{(j)}|}{1 + \lambda \delta^{(j)}} \, \epsilon.$$

This inequality shows that the contribution is jointly suppressed by (i) the factor $\epsilon$, which reflects the small causal effect, and (ii) the denominator $1 + \lambda \delta^{(j)}$, which further reduces unstable attributions. In particular,

$$\lim_{\epsilon \to 0} C_j = 0, \qquad \lim_{\delta^{(j)} \to \infty} C_j = 0.$$

Thus, spurious features with small output sensitivity are guaranteed to have negligible influence.

**Conclusion.** In both scenarios, either because the deviation score is above a threshold ($\delta^{(j)} \geq \Delta_{\min}$) or because the feature's effect on the output is negligible ($|\Delta M(x; j)| \leq \epsilon$), i.e., the refined attribution is bounded above and its contribution to $F_{\mathrm{RBP}}$ vanishes in the limit.

More concretely:

$$\tilde{a}^{(j)} \leq \frac{a^{(j)}}{1 + \lambda \Delta_{\min}} \quad \text{or} \quad C_j \leq \frac{|a^{(j)}|}{1 + \lambda \delta^{(j)}} \, \epsilon.$$

Therefore, spurious features are consistently down-weighted and rendered negligible in the refined faithfulness measure, establishing precisely the claim of Theorem 4.2. $\square$

# F   IMPLEMENTATION DETAILS

In this section, we describe the implementation details for the experiments conducted across tabular, image, and text data domains. We outline dataset generation, model configurations, explanation methods, and key experimental setups.

## F.1   TABULAR DATA EXPERIMENTS

**Dataset:** We used a synthetic regression dataset with six features. One spurious feature $\tilde{x}_3$ was injected, defined as $\tilde{x}_3 = \psi M(\mathbf{x}) + \varepsilon$, where $\psi$ controls the spurious correlation strength, and $\varepsilon \sim \mathcal{N}(0, \sigma^2)$.

**Models:** We evaluated three models:

- Random Forest (RF) with 100 trees.
- Support Vector Machine (SVM) with RBF kernel ($\gamma = 0.1$, $C = 1.0$).
- Multi-Layer Perceptron (MLP) with 2 hidden layers of size [128, 64].

**Explanation Methods:** Four post-hoc methods were applied: SHAP, LIME, Integrated Gradients (IG), and Occlusion. The explanations were evaluated under both normal and spurious conditions using the following metrics:

- Reliance on Outputs ($R$), Explanation Faithfulness ($F$), Inversion Score ($IS$), and Alignment ($A$).
- $\Delta IS$, measuring susceptibility to spurious correlations.

**Framework Implementation:** The Reproduce-by-Poking (RBP) framework applied perturbations to each feature and computed deviations to refine attributions. Ablation studies were performed by varying the number of spurious features, the strength $\psi$, and the number of perturbations.

## F.2 Image Data Experiments

**Dataset:** Shape classification was conducted on a synthetic dataset where images contained geometric shapes (e.g., circles, squares) with injected spurious distractor pixels in the top-left corner of images labeled as '1'.

**Models:** We used a CNN with 2 convolutional layers followed by 2 fully connected layers for the shape classification task.

**Explanation Methods:** We applied Integrated Gradients, Occlusion, Shapley Value Sampling, and LIME to generate visual attributions. These methods were evaluated using:

- $R$, $F$, $IS$, and alignment scores with ground-truth bounding boxes.
- Case studies to assess explanatory inversion on real-world image data.

## F.3 Text Data Experiments

**Dataset:** We designed a keyword-based classification task where keywords were embedded in sentences. During inference, spurious distractor tokens highly correlated with the output labels were injected.

**Models:** A pre-trained TinyBERT model was fine-tuned for the classification task.

**Explanation Methods:** We applied SHAP, LIME, and Integrated Gradients to generate token-level attributions, which were evaluated based on:

- $R$, $F$, $IS$, and alignment with ground-truth keywords.
- The impact of varying numbers of distractor tokens and perturbation magnitudes.

We did not use Occlusion for text tasks, since it's usually used for tabular or image data.

## F.4 Reproduce-by-Poking (RBP) Parameters

**Perturbation Process:** Features were perturbed three times per sample, with noise drawn from $\mathcal{N}(0, \sigma^2)$. Perturbation noise magnitudes ranged from $1\%$ to $9\%$ of feature values in sensitivity experiments.

**Hyperparameters:**

- The scaling factor $\lambda$ was set to 0.1 to refine attributions based on perturbation deviations.
- The power parameter $p$ in the Inversion Score was set to 2.

**Computational Resources:** Experiments were run on a machine with an NVIDIA RTX 3090 GPU and 64 GB of RAM. The code was implemented in Python using PyTorch, scikit-learn, and Captum for explanation generation.

## F.5 Explanation Methods

For explanations, we use the Captum library. We used KernelSHAP (with 1000 samples and background of 100 randomly sampled points), LIME (with default kernel and 500 perturbed samples), Integrated Gradients (baseline = zero vector, 50 integration steps), and Occlusion (sliding window of 1 feature at a time).

## F.6 Task Models

For tabular tasks, we used three models: Random Forest (with 100 estimators and max depth 5), Support Vector Regression with RBF kernel (C=1.0, gamma='scale'), and a 3-layer MLP (hidden sizes: [128, 64, 32], ReLU activation, trained with Adam for at most 20 epochs, learning rate 0.001).

For image tasks, we used a CNN with two convolutional layers (32 and 64 filters, kernel size 3, followed by max-pooling and 2 FC layers), trained on synthetic shape datasets. For text, we fine-tuned TinyBERT (2-layer, 312 hidden size) on binary keyword classification using HuggingFace's

Transformers library. Explanations were applied over the input embeddings with the same parameter settings as above.

To ensure fairness, all explanation methods shared the same perturbation or masking strategy per domain (e.g., pixel zeroing for images, word replacement for text, feature masking for tabular). Additional hyperparameters and training logs are provided in Appendix G.

## G  TRAINING DYNAMICS.

When training the deep models for evaluating explanation methods, we use the common practice for training, validation, and test steps. The training dynamics are shown in Figure 7. As observed, the task models have reasonable performance for evaluations.

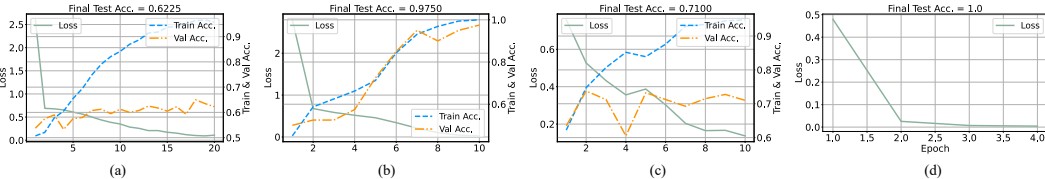

(a)  (b)  (c)  (d)

Figure 7: Train dynamics of (a) 2-layer CNN on CIFAR, (b) 2-layer CNN on synthesized shape classification dataset, (c) ResNet-18 on CIFAR, and (d) TinyBERT on synthesized keyword-based classification dataset.

## H  EXPLANATORY INVERSION IN SIMPLE LINEAR RELATIONSHIPS

Table 2: Results on Tabular data synthesized with a simple linear relationship. Scores are averaged with runs on 3 random seeds and reported as percentage (%).

| Model | Explanation | Baseline | | | Spurious Baseline | | |
|---|---|---|---|---|---|---|---|
| | | $R (\downarrow)$ | $F (\uparrow)$ | IS $(\downarrow)$ | $R (\downarrow)$ | $F (\uparrow)$ | IS $(\downarrow)$ |
| **Tabular (Linear relationship:** $y = x^{(1)} + x^{(2)} + \epsilon$**)** | | | | | | | |
| Random | SHAP | 0 | 100.0 | 0 | 0 | 100.0 | 0 |
| Forest | LIME | 0 | 100.0 | 0 | 0 | 100.0 | 0 |
| Linear Regression | SHAP | 0 | 100.0 | 0 | 0 | 100.0 | 0 |
| | LIME | 0 | 100.0 | 0 | 0 | 100.0 | 0 |
| SVM | SHAP | 0 | 100.0 | 0 | 0 | 100.0 | 0 |
| | LIME | 0 | 100.0 | 0 | 0 | 100.0 | 0 |
| MLP | SHAP | 0 | 98.5 | 1.1 | 0 | 98.0 | 1.4 |
| | LIME | 0 | 99.8 | 0.2 | 0 | 97.3 | 1.9 |
| | IG | 0 | 98.4 | 1.1 | 0 | 97.5 | 1.8 |
| | Occlusion | 0 | 99.1 | 0.6 | 0 | 98.0 | 1.4 |

Table 2 presents the results of explanatory inversion under a simple linear relationship between features and output, defined by $y = x^{(1)} + x^{(2)} + \varepsilon$. In this setup, we observe the following key trends:

1. **Absence of Inversion in Most Models:** For models such as Random Forest, Linear Regression, and SVM, both the reliance on outputs ($R$) and the Inversion Score ($IS$) are zero, indicating that these models produce faithful explanations without signs of explanatory inversion. The post-hoc methods (SHAP, LIME) provide explanations that align perfectly with the linear feature-output relationship, regardless of the presence of spurious features.

2. **Minimal Inversion for Neural Models:** In contrast, MLP-torch exhibits minor signs of explanatory inversion. While $R$ remains zero, the faithfulness score ($F$) decreases slightly under both baseline and spurious conditions. This leads to a small increase in $IS$ (up to 1.9 for LIME), showing that even in simple linear settings, neural models may introduce subtle non-linear artifacts that affect post-hoc explanations.

3. **Spurious Features Have Negligible Effect:** Across all methods and models, the spurious baseline results are virtually identical to the normal baseline. This highlights that explanatory inversion is strongly mitigated when the underlying feature-output relationship is linear, as post-hoc methods can accurately reflect the true feature contributions without being misled by spurious correlations.

Overall, these results suggest that explanatory inversion becomes less problematic when the model's decision function is simple and linear. This aligns with the theoretical expectation that explanatory inversion is exacerbated by complex, non-linear relationships and spurious influences.

## I   LIMITATIONS

While this work introduces a novel framework (IQ) for quantifying explanatory inversion and a method (RBP) for its mitigation, several limitations warrant consideration. The RBP method, involving forward perturbation checks, may introduce additional computational overhead, particularly for highly complex models or large datasets; its scalability for real-time applications remains an area for future optimization. Our empirical validation, though covering tabular, image, and text domains, primarily focuses on four general post-hoc explanation methods, and the evaluation of inversion with synthetic spurious feature injection might not capture the full spectrum of complex, real-world spurious correlations. Finally, the proposed RBP enhancement and IQ framework involve certain hyperparameters (*e.g.*, for perturbation and scoring sensitivity ) that may require careful tuning for optimal performance across diverse scenarios. Additionally, if not carefully configured (e.g., perturbation magnitude too large), it might distort feature context (especially in images or language). We mitigate this with small, localized perturbations and observe stable performance (see Sec. 5.4).

## J   BROADER IMPACT

This research has several broader implications for the field of explainable AI (XAI) and its application. By identifying and quantifying "explanatory inversion", our work can significantly contribute to developing more trustworthy and reliable AI systems, which is crucial for their adoption in high-stakes domains such as healthcare and finance. The proposed Inversion Quantification (IQ) framework and the Reproduce-by-Poking (RBP) method offer practical tools for researchers and practitioners to better assess and enhance the faithfulness of explanations, moving beyond mere justification of outputs. Ultimately, by fostering a deeper understanding of how explanations can sometimes mislead, this work encourages more critical evaluation of XAI techniques and promotes the development of methods that genuinely reflect a model's decision-making process, contributing to more robust and ethical AI.

## K   CASE STUDIES

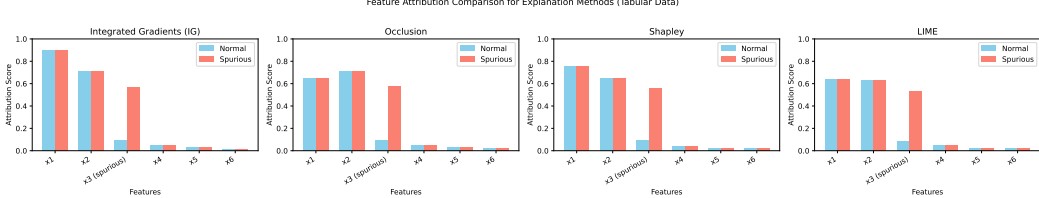

Figure 8: Feature attribution comparison for the multi-feature regression on tabular data across four explanation methods (Integrated Gradients, Occlusion, Shapley, and LIME) for tabular data. The bar charts display attributions for features $x_1$ to $x_6$ under both normal and spurious conditions. The spurious feature $x_3$ exhibits higher attribution under the spurious scenario, indicating a shift in explanation focus across all methods. While $x_1$ and $x_2$ maintain high relevance in normal conditions, the presence of a spurious correlation affects feature prioritization. This pattern demonstrates the potential susceptibility of explanation methods to spurious features.

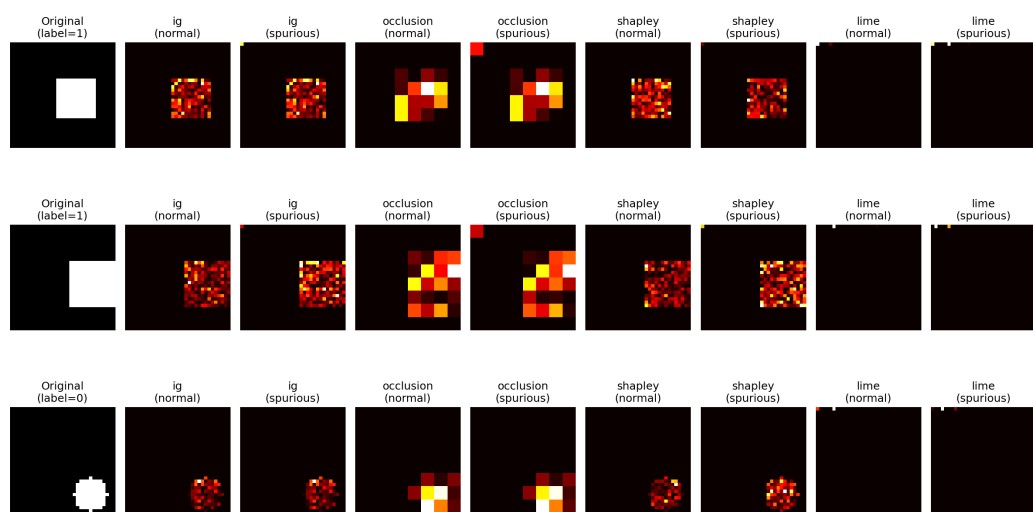

Figure 9: Visualization of feature attributions for the image classification task under both normal and spurious scenarios. Each row shows a different input image along with explanations generated by four post-hoc methods: Integrated Gradients (IG), Occlusion, Shapley Value Sampling, and LIME. Columns compare the attributions under normal conditions (middle) and spurious conditions (right). The spurious condition introduces a bright distractor pixel in the top-left corner, which shifts attributions toward the irrelevant region in several cases. The desired focus is highlighted by strong activations in relevant areas (e.g., shapes), while spurious influence results in increased attribution near the injected pixel.

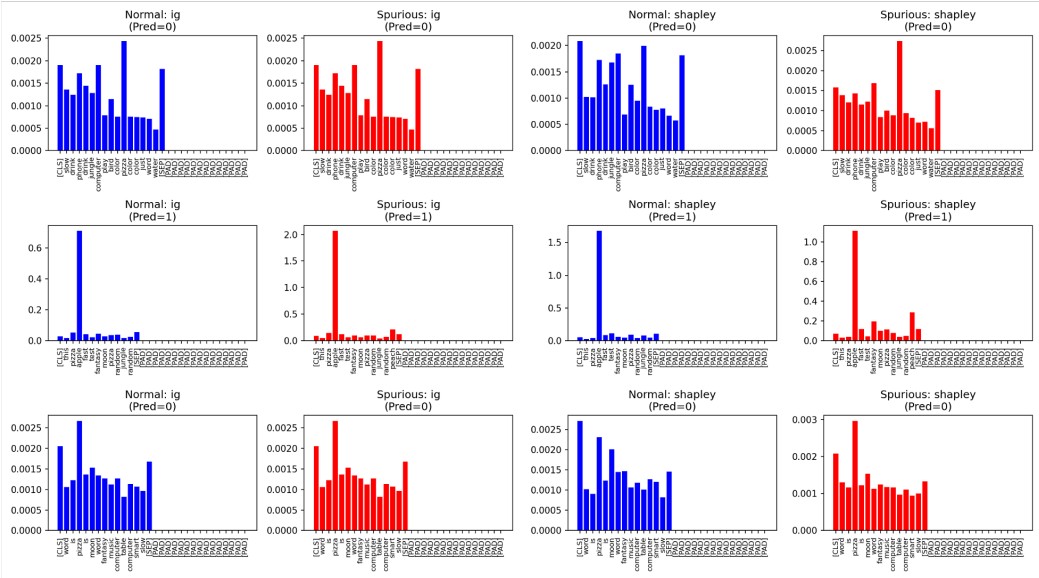

Figure 10: Feature attribution comparison on the text classification task under both normal and spurious conditions using two explanation methods: Integrated Gradients (IG) and Shapley Value Sampling. The horizontal axis shows tokenized input words, while the vertical axis represents attribution scores. The top rows illustrate results when the predicted label is 0, and the bottom rows show predictions of 1. Under spurious conditions (right panels), a spurious token (e.g., `peach`) affects the attribution scores, leading to a higher emphasis on non-relevant tokens. In contrast, under normal conditions (left panels), relevant tokens such as `apple` and `banana` maintain high attribution. This demonstrates the impact of spurious correlations on explanation consistency.

## L   RESULTS ON CIFAR DATASET WITH RESNET-18

We conduct an experiment on the CIFAR dataset. We select 2 classes ("cat" = 0, "dog" = 1). We randomly sample 500 instances from the training set and 200 instances from the validation and test set. The model is trained with a classic image classification pipeline. Explanatory inversions can be observed on certain samples.

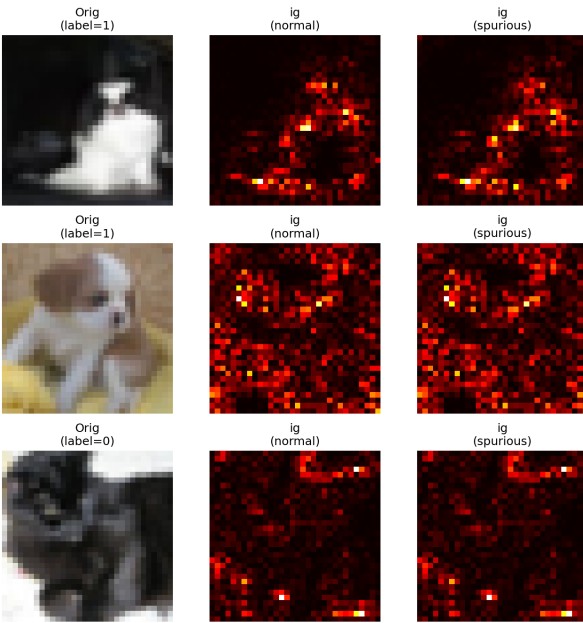

Figure 11: Case Studies on CIFAR

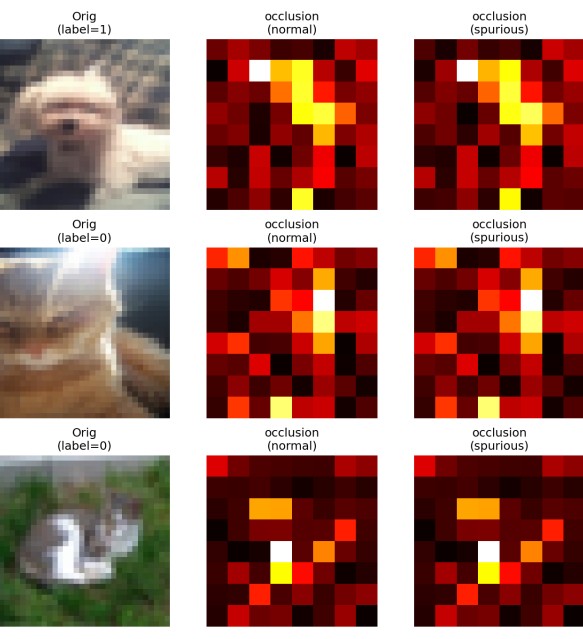

Figure 12: Case Studies on CIFAR

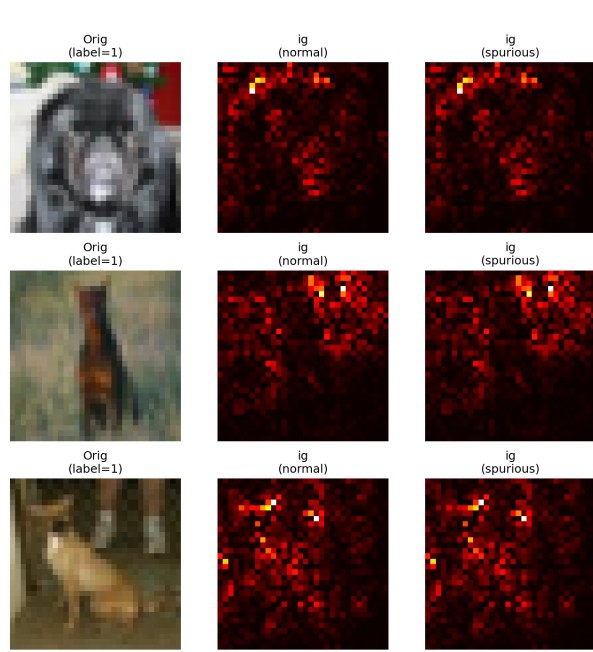

Figure 13: Case Studies on CIFAR

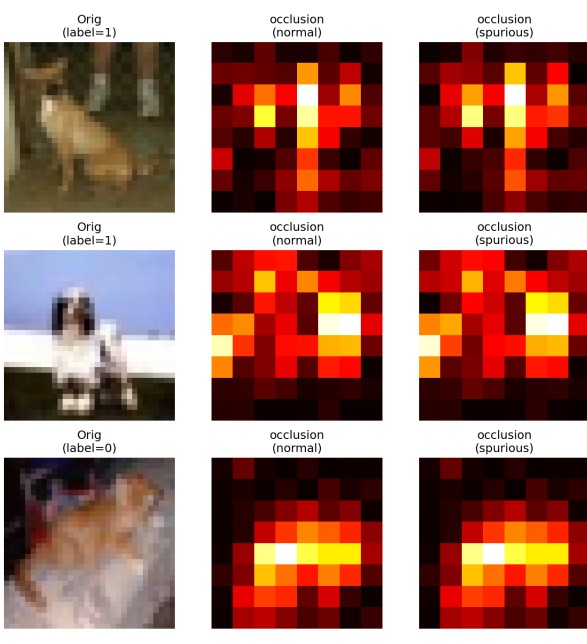

Figure 14: Case Studies on CIFAR

## M  CODE

Code is provided in the supplementary material.

