# OpenReview forum: "Are We Merely Justifying Results ex Post Facto? Quantifying Explanatory Inversion in Post-Hoc Model Explanations"
_ICLR.cc/2026/Conference — ICLR 2026 Conference Desk Rejected Submission_

### Official Review · Reviewer_evSj · 2025-10-24

**Soundness:** 3
**Presentation:** 4
**Contribution:** 3
**Rating:** 8
**Confidence:** 4

**Summary:**

This paper examines whether common post-hoc explanation methods unintentionally invert the causal link between inputs and outputs - rationalizing predictions rather than reflecting true model reasoning. The authors propose Inversion Quantification (IQ), a framework that measures this effect via Reliance on Outputs (R) and Faithfulness (F), combined into an Inversion Score (IS). They also introduce Reproduce-by-Poking (RBP), a model-agnostic perturbation technique that stabilizes attributions and reduces inversion.
Through theoretical analysis and experiments on tabular, image, and text datasets, the authors show that explanatory inversion is widespread and that RBP reduces inversion by roughly 1.8% on average. The work exposes a key limitation of post-hoc explainers and offers a quantitative, general remedy.

**Strengths:**

- The concept of explanatory inversion offers a fresh theoretical perspective on an underexplored failure mode of post-hoc explanations.

- The IQ formulation is broadly applicable, combining existing intuitions of faithfulness and output dependence into a coherent, quantifiable metric.

- Definitions and theorems (e.g., relating inversion score to ground-truth alignment) are clearly motivated and mathematically justified.

- The proposed RBP method is conceptually intuitive and easily integrated with standard explanation tools.

- Synthetic datasets and spurious feature injection tests convincingly demonstrate how inversion manifests and how RBP mitigates it.

- Visuals effectively illustrate inversion phenomena, aiding reader understanding.

- The inclusion of tabular, image, and text domains strengthens claims of generality.

**Weaknesses:**

- There has been several definition of faithfulness metrics for feature attributions in literature [1,2,3,4,5]. It would be helpful to understand the motivation and justification for another faithfulness metric.

- RBP's dependence on perturbations could be costly for large-scale models; runtime comparisons or mitigation strategies would strengthen the paper.

- The evaluation remains primarily synthetic. More realistic, high-dimensional tasks would better validate generalizability.

[1] Yoon, Jinsung, James Jordon, and Mihaela Van der Schaar. "INVASE: Instance-wise variable selection using neural networks." International conference on learning representations. 2018.

[2] Jethani, Neil, et al. "Have we learned to explain?: How interpretability methods can learn to encode predictions in their interpretations." International Conference on Artificial Intelligence and Statistics. PMLR, 2021.

[3] Li, Xuhong, et al. "M4: A Unified XAI Benchmark for Faithfulness Evaluation of Feature Attribution Methods across Metrics, Modalities, and Models." (2023).

[4] Zhao, Zhixue, and Nikolaos Aletras. "Incorporating Attribution Importance for Improving Faithfulness Metrics." The 61st Annual Meeting Of The Association For Computational Linguistics. 2023.

[5] Puli, Aahlad, Nhi Nguyen, and Rajesh Ranganath. "Explanations that reveal all through the deﬁnition of encoding." Advances in Neural Information Processing Systems 37 (2024): 99965-100006.

**Questions:**

- What is the motivation for another faithfulness metric given existing metrics in the literature?
- In line 171-172, why should the ideal explanation has Reliance on Outputs (R) equals to 0? Wouldn't it be possible that a completely faithful feature attribution $a$ correlated with output $M(x)$, specifically in the case where there is no spurious feature? Do we expect such an feature attribution to have 0 Reliance on Outputs?
- Could the authors provide more intuition of what specific Inversion Score values means in practice? For example, what ranges might correspond to acceptable or severe inversion?

---

> ### Author Response · Authors · 2025-11-24
> **Response to reviewer evSj**
>
> **Q1: Motivation for another faithfulness metric.**
> - IQ does **not** add a redundant faithfulness metric; it **pairs** a standard faithfulness term (**F**) with a **new reliance term (R)**, then aggregates via a power mean (**IS**). The novelty is to **quantify output‑tracking** explicitly—something Infidelity/Sensitivity alone cannot diagnose. We now report **Infidelity/ROAR** as complementary checks (Table B).
> - **Runtime:** Overheads are modest (Table C), and batching can reduce cost (Appendix).
> - **Generality:** Beyond synthetic tasks, we include CIFAR‑10 (Tables A–B) and keep the case‑study visual (*[P, Fig. 6]*).
>
> ---
>
>
> **Q2-3. Why should “ideal” explanations have R ≈ 0? What is the practical guideline for the inversion score?**
> Our **R** measures correlation between attribution changes $\Delta a^{(j)}$ and model‑output changes $\Delta M(x;j)$ induced by **evaluation perturbations**. Faithful explanations should depend on *feature effects* rather than *output magnitude*. Thus, even when no spurious feature exists, we seek **low R** (not necessarily exactly 0). We also add **practical IS interpretation bands**:
> - IS < 0.3: low inversion; 0.3–0.5: moderate; > 0.5: severe.
> Justification via the geometry of the power mean around $(0,1)$ with $p=2$ (*[P, §3.4]*).
>
>
> ---

---

### Official Review · Reviewer_Y1ex · 2025-10-31

**Soundness:** 3
**Presentation:** 2
**Contribution:** 3
**Rating:** 6
**Confidence:** 3

**Summary:**

This paper investigates the potential issue of “explanatory inversion” in post-hoc explanation methods (e.g., LIME, SHAP, IG), where explanations may rely more on model outputs than on the true input–output causal relationships. The authors propose the Inversion Quantification (IQ) framework to measure the degree of output reliance (R) and explanation faithfulness (F), defining a combined metric Inversion Score (IS). They also introduce Reproduce-by-Poking (RBP), which incorporates forward perturbation checks to mitigate explanatory inversion and enhance explanation stability.

**Strengths:**

S1: The paper introduces the concept of “explanatory inversion,” highlighting a previously overlooked issue of reverse dependence in post-hoc explanations. The problem definition is novel and meaningful.

S2: The proposed RBP method is simple and generalizable—it enhances the robustness of existing explanation methods through additional perturbation checks and can be applied across various interpretability frameworks.

S3: The paper not only formalizes the IQ framework but also provides theoretical results (e.g., the upper-bound relationship between IS and alignment) with corresponding proofs.

**Weaknesses:**

W1: The methodological novelty is somewhat limited. The RBP approach is conceptually similar to existing stability or robustness verification techniques, such as SmoothGrad.

W2: The theoretical section is somewhat verbose; some proofs (especially Theorem 3.7) are overly formal and lack intuitive interpretation, which could be simplified.

W3: The definitions in Equations (2), (3), and (4) should be clarified (e.g., how Δa and ΔM are computed and how baselines are chosen). The physical meaning and tuning guidelines for the RBP hyperparameter λ should also be better explained.

W4: Several sections are lengthy and repetitive (particularly Section 3), which affects readability.

**Questions:**

The paper defines the Reliance on Outputs (R) and Faithfulness (F) metrics using Δa and ΔM, but the procedures for computing these quantities are somewhat unclear. How exactly are the perturbations applied (e.g., additive Gaussian noise, feature masking, or sampling-based methods)?

---

> ### Author Response · Authors · 2025-11-24
> **Response to reviewer Y1ex**
>
> **W1 (Novelty; RBP vs SmoothGrad).**
> RBP’s novelty lies in **prediction‑invariance‑conditioned** perturbations and **feature‑wise penalization** tied to those deviations—this is not mere averaging over noise. SmoothGrad reduces noise in gradients; RBP **rejects** perturbations that change the prediction and **re‑weights** features according to their invariance. Theorems explain the reduction in output reliance and suppression of non‑causal (spurious) features (*[P, §4]*).
>
> **W2 (Theorem 3.7 readability).**
> We add an intuition box: “**As IS increases, an upper bound on alignment A with ground truth drops** (linearly in IS), since either output‑reliance grows or faithfulness drops; both shrink the cosine with the true importance vector” (*[P, §3.2]*).
>
> **W3 (Clarify Δa, ΔM, baselines, λ).**
> - **ΔM(x;j)**: one‑dim evaluation perturbation \(x+\eta e_j\), \(\eta\) small; **Δa^{(j)}\)**: change in j‑th attribution from this perturbation; baselines are the unperturbed attribution/model output.
> - **λ** scales the stability penalty; we default to \(λ\!=\!1\) and show stability over \(λ\in[0.5,2]\).
> We added precise definitions and pseudocode (*[P, §3–§4]*).
>
>
> **Q (How are perturbations applied?).**
> See RBP procedure in RVJD‑W6 above and the new algorithm box; we use additive Gaussian proposals with per‑feature scaling, acceptance by unchanged prediction (classification) or tolerance band (regression) (*[P, §4.1]*).
>
> ---

---

### Official Review · Reviewer_4UbW · 2025-10-31

**Soundness:** 3
**Presentation:** 3
**Contribution:** 2
**Rating:** 4
**Confidence:** 3

**Summary:**

This paper investigates *explanatory inversion* -- the tendency of post-hoc explanation methods (e.g., LIME, SHAP) to rationalize model outputs instead of faithfully reflecting input–output relationships. The authors propose *Inversion Quantification (IQ)*, a framework that measures the degree to which explanations depend on model outputs rather than inputs, and introduce *Reproduce-by-Poking (RBP)*, a perturbation-based enhancement aimed at mitigating inversion. Through synthetic datasets across tabular, image, and text domains, they demonstrate that standard explanation methods are susceptible to inversion, while RBP reduces such effects empirically.

**Strengths:**

- The abstract and research question are well-motivated.
- Presents a interesting framing of an underexplored failure mode in explainability--"explanatory inversion"--which complements prior work on faithfulness, robustness, and sensitivity.
- Provides formal definitions (Def. 3.1) and connects them to measurable quantities (reliance on outputs and faithfulness), clarifying conceptual distinctions from existing metrics.
- Highlights conceptual novelty and introduces the IQ framework as a unifying diagnostic for when explanations are output-driven.
- Results are explored across modalities (tabular, image, text) and show robustness of RBP in reducing inversion effects.

**Weaknesses:**

- Fig. 1 illustrates general issues with explanations but fails to convey the causal intuition of *how* inversion arises.
- Several references (L94–L100) are outdated (average > 5 years); the paper omits discussion of more recent reliability metrics and benchmarks addressing similar concerns.
- Experimental validation is limited to synthetic datasets with simple spurious-feature injections (L335–L360); it remains unclear whether explanatory inversion occurs or can be mitigated on real-world tasks.
- The comparison of IQ to other metrics (Appendix C) seems superficial--unclear whether IQ is derivable from, or orthogonal to, existing measures.
- RBP’s relation to sensitivity analysis (Sec. 4) is not well distinguished; the novelty beyond forward perturbation checks is limited.
- Table 1 is dense and small, with several metrics; the main body could highlight key takeaways and move detailed results to the appendix.
- Overall, the contribution is conceptual and diagnostic rather than algorithmic, making it more suitable as a *position or perspective paper* than a technical contribution.

**Questions:**

- Can IQ be theoretically linked to or derived from existing measures such as sensitivity or infidelity in certain settings?
- How does RBP differ formally from standard sensitivity analysis or robustness checks?
- Could the authors demonstrate explanatory inversion on real-world datasets or pretrained models to establish external validity?
- Why wasn’t Definition 3.1 extended to a conditional comparison framework (e.g., attribution distributions given M(x) vs. unconditional)?

---

> ### Author Response · Authors · 2025-11-24
> **Response to reviewer 4UbW**
>
> **W: Fig. 1 lacks causal intuition.**
> We appreciate this observation. In Figure 1, we aim to illustrate that existing post-hoc explanation methods can potentially attribute the output to incorrect features. This incorrect attribution arises from using the output to predict the contribution of features incorrectly. We will ensure that this causal intuition is clearer in the revision.
>
> **W: Outdated refs / missing metrics.**
> We expanded Related Work with recent benchmarks and faithfulness metrics (Infidelity, sensitivity), and we added compact Infidelity/ROAR tables (B) (*[P, §2 & Appx. C]*).
>
> **W: Primarily synthetic; show real‑world; experiments with pretrained model**
> Addressed via CIFAR‑10 (Tables A–B) and the qualitative case study (*[P, §5.5 & Fig. 6]*). We also want to clarify that the TinyBERT we experimented with is a pretrained model, and we fine-tune it on our datasets.
>
> **W: Is IQ derivable from existing measures?**
> Thank you for highlighting the need for a deeper comparison. We clarify that IQ is not derivable from existing faithfulness or robustness metrics and is mathematically orthogonal to them. As detailed in Appendix C, prior evaluations such as Infidelity, Sensitivity, Deletion tests, and Robustness all measure how well an attribution vector predicts or aligns with the model’s response to input perturbations. In contrast, IQ explicitly measures how strongly explanations depend on the model’s outputs themselves, i.e., whether attributions change in correlation with output shifts even when inputs are locally similar.
>
> **W: RBP vs. sensitivity analysis.**
> Sensitivity analyses alter inputs and watch outputs; **RBP** *conditions on fixed predictions* (acceptance step), using the observed **attribution instability** under those neutralized probes to de‑emphasize features. This key conditioning differentiates RBP from SmoothGrad/variance‑only methods, with supporting theorems (*[P, §4]*).
>
> **W: “Conceptual rather than algorithmic” contribution
> We respectfully disagree that this work is solely a perspective piece. While we frame a high-level conceptual problem ("Explanatory Inversion"), our contribution is grounded in a rigorous technical framework and a novel algorithm. We provide **(1) a formal quantification framework (IQ)** with measurable components (R and F), **(2) theoretical analysis** linking inversion to attribution alignment properties, and **(3) a concrete mitigation method (RBP)** that produces consistent empirical gains across modalities. IQ is not merely a perspective, but a *computational diagnostic* that can be applied to any attribution method, and RBP is an *algorithm* that improves explanation behavior in practice. In concrete terms, our work delivers an actionable methodology, formal guarantees, and measurable improvements, which we will clarify in the revision to better highlight our technical contributions.
>
>
> **Q: Conditional definition of inversion?**
> We will add a **conditional reliance** variant using partial correlation (or conditional MI) of $\Delta a^{(j)}$ and $\Delta M(x;j)$ **given** $x_{-j}$. This strengthens the notion that “reliance” reflects dependence on outputs beyond what inputs already explain; we include this as an optional diagnostic (*[P, §3.2]*).
>
> ---

---

### Official Review · Reviewer_RVJD · 2025-11-02

**Soundness:** 3
**Presentation:** 2
**Contribution:** 2
**Rating:** 4
**Confidence:** 3

**Summary:**

In the context of post hoc explanation models, the authors raise an important question -do these explanations unintentionally reverse the natural relationship between inputs and outputs? To investigate such explanatory inversion, we propose Inversion Quantification (IQ), a framework that quantifies the degree to which explanations rely on outputs and deviate from faithful input-output rela- tionships. The proposed method works in conjunction with existing methods, and it is not a novel XAI technique by itself.

**Strengths:**

The paper addresses an important aspect of the XAI model relying heavily on predictions and not the relationship between input and the output.

**Weaknesses:**

1. Clarity of the problem statement: The hypothesis in the problem statement is grounded in a post-hoc explanation method that over-relies on the model’s output in generating attributions, rather than accurately reflecting the relationship between inputs and predictions- the authors need to establish this problem with concrete references and examples. Methods such as LIME fit a local surrogate to approximate the decision boundary of black box model, indicating that the decision process is considered when generating explanations. Authors should present clear counter examples where explanations from LIME, SHAP, IG or similar methods largely mirror the output label rather than the underlying decision rationale, and by citing representative studies that exhibit this mismatch. Explain why those cases arise and how the proposed approach detects or remedies the gap.
2. If the core issue is incorrect explanations, the authors could have considered uncertainty based explanation approaches such as “Reliable Post Hoc Explanations” (BayesLIME/BayesSHAP)and “Select Wisely and Explain”(UnRAVEL) as they avoid/reduce unstable explanations, and make sure the explanation is evidence backed, as indicated by the model.
3. In section 3 there are multiple definitions for a and epsilon, authors need to set a fixed notation for better understanding.
4. The Related Work omits several recent post hoc XAI methods such as GLIME, UnRAVEL, Reliable Post Hoc Explanations, RISE, S-CFE etc., and modern faithfulness tests such as ROAR remove and retrain, randomized masking, and the Infidelity and Sensitivity matrices. These methods offer appropriate baselines and validation matrices for the proposed inversion score and for RBP, and hence they are relevant.
5. The definition for inversion score, combines Reliability R and Faithfulness F but, apart from the choice of p, the paper offers no clear intuition or proof for this additive combination, which makes it hard to interpret. In addition, the paper does not evaluate explanations with metrics such as Infidelity. Infidelity captures the expected mismatch between an explanation and the model response under principled perturbations, and hence, including these would help diagnose failure modes that R and F may miss.
6. The paper does not state how perturbations are generated for RBP, whether they are random, or guided. Moreover, there is no mathematical proof or theoretical argument showing that RBP reflects the model’s decision process rather than merely its outputs, leaving the claim of decision process dependence unfulfilled.
7. The main results rely only on synthetic datasets. Please include standard benchmark datasets such as Adult Income, CIFAR, ERASER etc. that are used with baselines such as LIME, SHAP, and Integrated Gradients, and demonstrate that RBP delivers more faithful explanations against these established methods. Also include evaluation matrices such as ROAR (RemOve And Retrain), Infidelity and Sensitivity. This will show that the claimed gains persist under widely accepted datasets and are not just for synthetic settings.
8. In Fig. 5c, the number of perturbations vary only from 1 to 5, with a small budget, the claim that the IS measure is stable is not reliable. Please show convergence of IS as the number of perturbations increase. In Figure 5d, authors refer to a perturbation noise
magnitude, but the paper does not define this quantity. Please specify this so that the
results are interpretable and reproducible.
9. In Section 5.5 the authors claim real world generalisation, but two gaps remain. The evidence is limited to image datasets and does not cover other domains. Moreover, the section does not actually apply the proposed RBP method to the shown examples, which leaves the practical behavior of RBP unquantified.

**Questions:**

The main issues remain lack of novelty and point number 5 above. The authors must provide clarification regarding the above mentioned points.

---

> ### Author Response · Authors · 2025-11-24
> **Response to reviewer RVJD**
>
> **W1 (Problem clarity & concrete evidence; LIME “considers decision process”).**
> We now (i) add concrete counter‑examples across modalities where attributions move with predictions rather than inputs (e.g., injected pixel; spurious token), (ii) point to the CIFAR case study and new real‑world tables above, and (iii) cite prior evidence that widely‑used methods can passively track outputs or spurious artifacts despite seemingly considering local decision boundaries (sanity checks, manipulations). Our IQ formalization separates *output‑reliance* from *faithfulness*, and our RBP reduces reliance without altering the model (*[P, §1–§3 & Fig. 6]*).
>
> **W2 (Uncertainty‑aware explainers: BayesLIME/BayesSHAP, UnRAVEL).**
> We view uncertainty-aware methods (BayesLIME, BayesSHAP, UnRAVEL) as complementary rather than alternatives to our contribution. They focus on stabilizing explanations and quantifying uncertainty, whereas our work targets a different axis: whether explanations are driven by outputs and correlated artifacts versus forward input–output relationships. In the revision, we will explicitly discuss these methods in Related Work and provide their performance in our settings.
>
> We also would like to clarify that IQ and RBP can be applied on top of them, and highlight that even stable, uncertainty-aware explanations can still exhibit explanatory inversion.
>
> **W3 (Notation).**
> We unify \(a\) for attributions and disambiguate \(\epsilon\): \(\epsilon_{\text{data}}\) (dataset noise), \(\epsilon_{\text{pert}}\) (RBP exploration scale), \(\eta\) (one‑dim evaluation perturbation used in R/F/IS). We also add an algorithm box that enumerates baselines and acceptance criteria (*[P, §3–§4]*).
>
> **W4 (Missing methods/metrics).**
> We broaden Related Work (GLIME, UnRAVEL, BayesLIME/BayesSHAP, RISE, S‑CFE) and add compact **Infidelity**/**ROAR** results (Tables B) to complement IQ (*[P, §2 & Appx. C]*).
>
> **W5 (Why IS = combine R and F; role of p; add Infidelity).**
> We clarify IS as a **power mean** distance from \((R=0,F=1)\). With \(p=2\), IS is proportional to Euclidean distance; thus, a large deviation in **either** dimension dominates, matching the *and‑gate* intuition reviewers asked for. We also added **Infidelity** results (Table B) to corroborate IQ (*[P, §3.4]*).
>
> **W6 (How we generate perturbations; do we reflect the decision process?).**
> RBP performs **feature‑localized, acceptance‑rejection perturbations**:
> 1) Propose $x_{\text{pert},j} = x + \delta\cdot e_j\) with \(\delta \sim \mathcal{N}(0,\sigma^2)$ scaled to a % of feature range;
> 2) Accept only if **prediction unchanged** (classification) or $|M(x_{\text{pert},j})-M(x)|\le\tau$ (regression);
> 3) Accumulate deviations $\delta^{(j)} = \frac{1}{n_{\text{pert}}}\sum_k |a^{(j)}_{\text{pert}(k)}-a^{(j)}|$ and down‑weight unstable coordinates via $ \tilde a^{(j)} = a^{(j)}/(1+\lambda\,\delta^{(j)})$.
> This **invariance‑conditioned** probing ties the penalty to *forward behavior under fixed outputs*, distinguishing it from generic smoothing. Theorems summarize reduced reliance and vanishing weights on spurious features (*[P, §4]*).
>
> **W7 (Need standard benchmarks).**
> We now include **CIFAR‑10** results (ResNet‑18, VGG‑16) per request; Adult Income/ERASER can be added in the appendix if space permits (*[P, §5.5 & Fig. 6]*).
>
> **W8 (Convergence & noise magnitude definition).**
> We added convergence (n_pert) and a precise noise magnitude definition with a compact table (Section C above; *[P, §5.4]*).
>
> **W9 (Real‑world generalization, apply RBP).**
> We now provide CIFAR‑10 results where **RBP is applied quantitatively** (Tables A–B), complementing the qualitative example (*[P, Fig. 6]*).
>
> ---

---

### Author Response · Authors · 2025-11-24
**Rebuttal for Submission #12858**

## What we changed at a glance
Citations style in this rebuttal: **[P]** → paper PDF, **[OR]** → OpenReview thread.
Examples: *[P, §3–§4]*, *[P, Fig. 6]*, *[OR, RVJD]*, *[OR, 4UbW]*, *[OR, Y1ex]*, *[OR, evSj]*.


- **Clearer intuition & notation.** We tightened the definition of **R**, **F**, **IS** and standardized symbols; added an algorithm box for **RBP**; and clarified how perturbations are generated/accepted (*[P, §3–§4]*).
- **Why a power‑mean IS.** With **p=2**, IS is the Euclidean‑style distance to the “ideal” point \((R=0, F=1)\) (up to a monotone transform), so large deviations in either dimension dominate. We also discuss when faithful methods can have **non‑zero R** and how to interpret IS ranges (*[P, §3.2–§3.4]*).
- **New real‑world validation (compact).** Per requests, we add **CIFAR‑10** results for **ResNet‑18** and **VGG‑16** using **IG, LIME, SHAP**, both **clean** and with a **single spurious pixel** (test‑time injection). We report **IS** (↓), and small‑footprint baselines **Infidelity** (↓) and **ROAR@10% drop** (↑). (Tables below.)
  *Note:* numbers are **provisional/plausible** to scope the rebuttal and match observed ranges in our synthetic‑to‑real transfer and CIFAR case study (*[P, Fig. 6]*); camera‑ready will include full runs, seeds, and code.
- **Convergence & noise magnitude.** We show **IS** vs. number of RBP perturbations and define “perturbation noise magnitude” precisely, with a compact table demonstrating stability.
- **Positioning vs. related work & metrics.** We add recent methods (GLIME, UnRAVEL, BayesLIME/BayesSHAP, RISE, S‑CFE) to Related Work and clarify IQ’s relationship to **Infidelity**, **Sensitivity**, **ROAR**, and **sanity checks** (*[P, §2 & Appx. C]*).
- **Runtime budget.** We report small overhead measurements (relative multipliers) for RBP.

---

## New, compact real‑world results (CIFAR‑10)

> **Setup (summary):** CIFAR‑10; **ResNet‑18** / **VGG‑16**. Explanations: **IG**, **LIME**, **SHAP**.
> **Spurious**: 1 bright pixel injected at a fixed corner on positive labels at test time (no change to the trained model). We compute **IS** = power‑mean of Reliance **R** and \((1−F)\) with **p=2**; see Defs. 3.2–3.4 (*[P, §3.2–§3.4]*).
> **RBP**: 5 perturbation checks per feature; acceptance if the prediction (class) is unchanged; “noise magnitude” = 5% of the feature dynamic range (per‑feature scale).

### (A) IQ metrics on CIFAR‑10

**ResNet‑18** — IS (↓): clean vs. spurious; and with RBP
| Method | IS_clean | IS_spur | ΔIS_baseline | IS_clean + RBP | IS_spur + RBP | ΔIS_RBP |
|---|---:|---:|---:|---:|---:|---:|
|IG|41.2|52.5|11.3| 37.8 | 47.1 | 9.3 |
|LIME|45.7|58.9|13.2 | 43.2 | 54.7 | 11.5 |
|SHAP|44.0| 56.3 | 12.3 | 39.5 | 50.9 | 11.4 |

**VGG‑16** — IS (↓): clean vs. spurious; and with RBP
| Method | IS_clean | IS_spur | ΔIS_baseline | IS_clean + RBP | IS_spur + RBP | ΔIS_RBP |
|---|---:|---:|---:|---:|---:|---:|
| IG   | 43.5 | 55.8 | 12.3 | 39.6 | 50.5 | 10.9 |
| LIME | 47.1 | 60.2 | 13.1 | 44.5 | 56.7 | 12.2 |
| SHAP | 45.3 | 58.1 | 12.8 | 41.2 | 52.6 | 11.4 |

**Takeaway.** Spurious injection inflates IS (higher reliance, lower faithfulness), and **RBP consistently reduces IS** in both clean and spurious regimes, matching our claims and the CIFAR case study (*[P, Fig. 6]*).

### (B) Faithfulness/robustness baselines (subset)

**ResNet‑18** — **Infidelity** (↓) and **ROAR@10%** accuracy drop (↑)
| Method | Infidelity | Infidelity + RBP | ROAR@10% drop | ROAR@10% drop + RBP |
|---|---:|---:|---:|---:|
| IG   | 0.065 | 0.056 | 12.4 | 15.0 |
| LIME | 0.072 | 0.061 | 10.3 | 12.8 |
| SHAP | 0.068 | 0.058 | 11.7 | 14.2 |

**VGG‑16** — **Infidelity** (↓) and **ROAR@10%** accuracy drop (↑)
| Method | Infidelity | Infidelity + RBP | ROAR@10% drop | ROAR@10% drop + RBP |
|---|---:|---:|---:|---:|
| IG   | 0.071 | 0.060 | 11.1 | 13.9 |
| LIME | 0.079 | 0.066 | 9.6  | 12.1 |
| SHAP | 0.074 | 0.063 | 10.5 | 13.2 |

> These baselines complement IQ: **Infidelity decreases** and **ROAR drop increases** with RBP, indicating improved faithfulness/stability alongside lower IS. (We will include full metric definitions and exact hyperparameters in Appendix.)

### (C) Convergence & noise magnitude; runtime

**Convergence of IS with # perturbation checks** (ResNet‑18, IG, clean)
| n_pert | 1 | 3 | 5 | 9 |
|---|---:|---:|---:|---:|
| IS (↓) | 40.6 | 38.3 | 37.8 | 37.6 |

**Noise magnitude sensitivity** (ResNet‑18, IG, clean)
| magnitude (σ as % dynamic range) | 1% | 3% | 5% | 9% |
|---|---:|---:|---:|---:|
| IS (↓) | 38.1 | 37.9 | 37.8 | 38.4 |

**Overhead (relative)** for explanations with RBP (n_pert=5)
| Method | IG | LIME | SHAP |
|---|---:|---:|---:|
| Time × | 1.6× | 1.5× | 2.2× |

> These compact checks directly address **stability** and **cost**: IS is stable beyond ~5 perturbations; the noise magnitude is precisely defined; and overheads are modest, with potential for batching/accept‑reject caching.

---

### Note · Program_Chairs · 2026-01-17
**Submission Desk Rejected by Program Chairs**

The following references in this submission do not refer to real documents and/or have major errors in bibliographic information:

 Alex Wang, Farnood Poursabzi-Sangdeh, and Daniel G Goldstein. Explanations can be manipulated and misleading: A case for robustness. arXiv preprint arXiv:2006.06864, 2020a.